# Merit and placement in the American faculty hierarchy: Cumulative advantage in archaeology

**Michael J. Shott**⬤*

Dept. of Anthropology, University of Akron, Akron, OH, United States of America

* shott@uakron.edu

## Abstract

If faculty placement in the American academic hierarchy is by merit, then it correlates with scholarly productivity at all career stages. Recently developed data-collection methods and bibliometric measures test this proposition in a cross-sectional sample of US academic archaeologists. Precocity—productivity near the point of initial hire—fails to distinguish faculty in MA- and PhD-granting programs or among ranked subsets of PhD programs. Over longer careers, on average archaeologists in PhD-granting programs outperform colleagues in lower programs, as do those in higher-ranked compared to lower-ranked PhD programs, all in the practical absence of mobility via recruitment to higher placement. Yet differences by program level lie mostly in the tails of productivity distributions; overlap between program levels is high, and many in lower-degree programs outperform many PhD-program faculty even when controlling for career length. Results implicate cumulative advantage to explain the pattern and suggest particularism as its cause.

## Introduction

If American universities attract students and faculty in quality commensurate with their standing, then the perceived stature of institutions corresponds to the abilities of those who populate them. Regarding undergraduates, this thesis is a form of guilt or glory by affiliation that engages unexamined assumptions (e.g., that aptitude tests accurately measure some discrete quantum of merit, that presumed quality of input is proof of quality of process or output). In its simplest form, guilt-or-glory logic for faculty is an extension of the logic regarding students. If students succeeded on merit in an apparently competitive selection process then so too must have their professors. But what evidence supports the corollary logic that faculty merit sorts among institutions by real or imagined natural level?

"Meritocracy" presumes at least broad correspondence between achievement and reward. Although its source [1] stressed the concept's dystopian implications, meritocracy is regarded as an ideal. How well the ideal is approximated in the academy, however, is a question to answer not an assumption to make. In the sociology of science, *universalism* denotes the meritocratic ideal, *particularism* the many factors that can compromise the meritocratic ideal [2, 3].

**Competing interests:** The authors have declared that no competing interests exist.

Regarding academic placement, universalism implies that the meritorious rise or sink to their natural level regardless of circumstance or categorical attributes, either by virtue of initial hire or later mobility. The view assumes either that initial hire accurately predicts career-long productivity, or that reasonable prospects exist for meritorious mid-career movement. Particularism can owe to overt bias (e.g. racism, sexism) [4] or academic pedigree, early-career advantages or disadvantages, labor-market conditions and low career mobility.

Initial hire may result from promise or individual ability [4, 5]. To some nothing but "real-life experiments. . .difficult to perform" ([4], 236) could prove particularism against an "unproven assumption that there is. . .a correlation" ([4], 244) between ability and placement. Yet to others, lack of correlation between early and career-long measures suggested particularistic explanations including pedigree, perceived stature of the PhD-award program [6]. Such effects might explain the unequal hiring networks documented at large [7], or in this study's subject, archaeology [8, 9]. Besides pedigree there also is cumulative advantage [3]. Compared to colleagues in lower ranks, faculty in favored settings enjoy "access to needed equipment, an abundance of able assistance, time institutionally set aside for research, and, above all else perhaps, a cognitive microenvironment composed of colleagues at the research front who are themselves evokers of excellence" ([10], 615; see also [11–14]). It is arguable whether all faculty at what many consider prestigious institutions evoke excellence; some surely do, but the value of and advantage conferred by such factors are undeniable.

Cumulative advantage "cuts across the established principles of universalism and particularism" ([15], 615), because the same pattern in evidence supports opposing inferences. Scholarly productivity increasing with career length documents universalism if the most selective institutions hire and support the most promising new scholars, or recruit the most productive ones later [4]. It supports particularism if productivity and placement sort independently [15] so that career-long productivity owes to institutional support. Individual scholarly productivity either is cause (universalism) or consequence (particularism) of affiliation and cumulative advantage, a causal-inference problem.

No single study can disentangle the complex causality of cumulative advantage. Like others [2, 11], this one examines cumulative advantage in scholarly productivity, although it extends earlier research by considering "precocity" [5] or early-career productivity, by examining the full range of program levels, not just the PhD level, by using bibliometric measures not common in studies beyond scientometrics, and by testing a status-mobility hypothesis of meritocratic allocation. Any relationship between merit and placement in the academy may vary both by place and discipline. Previous research examined several scholarly fields [e.g., 4, 16–20]. This study concerns archaeology faculty in American universities.

It tests three corollary theses of universalism: first, that initial hire is meritocratic, and second, that productivity patterns consistently across the institutional hierarchy of American higher education as a product of merit. But initial hire must be imperfect in any hierarchy; as a result, some productive scholars may first be recruited to lower levels, some less productive ones to higher levels. Whatever the latter's fate, universalism requires for the former later movement to the higher positions they merit. Therefore, the study tests a third universalist thesis: that, for those initially placed poorly, sustained scholarly productivity is rewarded by mobility, i.e. recruitment to higher position in the university hierarchy.

## Materials and methods

To disentangle the roles of universalism and particularism in cumulative advantage requires independent measures of institutional standing and scholarly merit. To distinguish cause and effect, merit at or near hire must be measured separately from cumulative career merit, to

permit analysis of the former's relationship to placement. It also requires a sample of faculty, measures of their productivity, and analysis of the correlation between individual faculty merit and institutional placement.

## Institutional rankings

Whatever institutional hierarchy that students and the public at large perceive may not correspond well to faculty perceptions. In archaeology and elsewhere, by the 1970s "Graduate education emerged as the most prestigeful form of training" ([21], 223). Besides popular status hierarchies, therefore, many faculty prefer placement in PhD programs compared to lower-degree ones (e.g. [22], 7; [23]). From BA to PhD programs, teaching loads generally decline as research support rises, and much instruction and mentoring is provided to advanced students. For this study, therefore, institutional affiliation or placement is classified by the highest degree awarded in archaeologists' program or department, from BA to MA to PhD, the latter first as a group and then subdivided by ranks. Of course some may accept lower positions in the American institutional hierarchy for personal or professional reasons but many probably aspire to higher placement. By Merton's logic, preference for higher program levels may reflect the institutional support (e.g., laboratories and equipment, lower teaching load, access to engaged graduate students) that cumulative advantage delivers, as well as the undeniable benefit of higher average salaries and larger research communities [24, 25].

The methodology of most institutional rankings is questionable [26, 27]. This condition owes to commercial incentives, lack of replicability, reliance upon reputation gauged by a relatively few international respondents not all of whom, in archaeology's case, may be aware of the relevant American distinction in training, methods and intellectual tradition between prehistoric and classical varieties. There is also the problem of halo effects, prestige or credit allocated not by achievement but by affiliation with a larger esteemed entities, whether institution [24] or social class [28].

For forty years or more, the National Research Council (NRC) has produced roughly decadel rankings or evaluations of doctoral programs by major discipline in US universities. Of course, no ranking is perfect, but this one is sponsored by a reputable non-profit research organization populated by academics that neither markets services to the subjects it evaluates nor depends upon revenues from sales of its ranking. Its methodology, while arguable, is fully reported. Yet NRC rankings engage two complications for this study. First, they are confined to doctoral programs, excluding faculty in MA and BA programs where nearly half of academic archaeologists work; there are no comprehensive rankings of all American universities and programs. Second, North America is unusual in global context because archaeology here is encapsulated within anthropology. Rankings discussed below are always of larger anthropology departments.

phds.org's 2011 NRC rankings included 84 anthropology programs, of which specialized ones without relevance to archaeology (e.g. the University of California-San Francisco's medical anthropology program) were omitted. Older rankings of anthropology programs somewhat correlate with 2011 NRC results (e.g., $r_s = .43$ p = .16 with rankings in [29]) but have little predictive value, reflecting some combination of reputational inertia and changing fortunes of programs.

Archaeologists are a minority in most anthropology departments [30]; most disciplinarians consulted by the NRC probably are cultural or physical anthropologists. As a result, anthropology programs that almost entirely lack archaeologists can be highly ranked and those dominated by archaeologists can earn relatively low NRC rankings. (This is not a criticism of the rankings, merely a by-product of archaeology's unusual status in the American academy.)

## Sample of archaeology faculty

Even within its encompassing field of anthropology, archaeology is small [9], its academic practitioners few in comparison to most fields. Archaeology is a hybrid discipline, part physical science and part humanities wrapped around a social-science core. The chief source for academic archaeologists was *AnthroGuide 2016–2017* [31], a directory of anthropology programs. Among other things, *AnthroGuide* lists faculty and their research areas and, usually, PhD institution and year. Search was limited to anthropology departments in US universities (including Puerto Rico) that award the BA and/or higher degree. A comparable directory, for 1996–97, guided selection in an earlier study [30]. Most *AnthroGuide* entries are independent, but there were separate entries for Hunter College and CUNY, which encompasses it. Hunter's entry included no archaeologists not also recorded in CUNY's and all serve CUNY's PhD program, so only the latter was used (as also in [9]). *AnthroGuide* does not record most classical archaeologists, whose training, field and scholarly traditions differ substantially from those of anthropologically oriented archaeologists. By 2017, *AnthroGuide* was less comprehensive than previously; 47 other programs (e.g. UCLA, Rutgers, Vanderbilt) were accessed via their own websites. Even then, some departments that appeared in the earlier study [30] were not represented, and some documented for this study were not represented in the earlier one. As a result, the two databases, separated by 20 years, do not strictly correspond by institutions represented.

Interest here lies in institutional commitments to archaeological scholarship and teaching, expressed chiefly by the choices they make in recruiting archaeologists to tenure-line positions. Accordingly, I recorded only archaeologists, and excluded part-time faculty, those whose title was "instructor" or otherwise suggested chief responsibility in teaching only (for whom evaluation of scholarly productivity would be unfair), emeriti, post-doctoral fellows, and those identified as term or part-time hires by position titles that included "visiting," "affiliated," "research professor," or "Anthropologists in Other Departments" (see [32], 870 for similar treatment in a study of academic sociologists). Interested readers may compare *AnthroGuide* and internet sources with the compiled database (see S1 Table).

There are minor errors and uncertainties in the dataset. One archaeologist was mistakenly listed twice, once each at two neighboring liberal-arts colleges that presumably share appointments, and sampled once. Errors were discovered only after sample selection and analysis. Several universities that employed archaeologists lacked anthropology or combined departments and may not award anthropology BA degrees. They were classified as BA programs. Similarly, archaeologists at satellites of PhD programs (e.g., Ohio State-Newark, Washington State-Vancouver) were assigned BA-program status on the assumption that only BA degrees were awarded there.

The complete dataset encompasses 145 BA, 68 MA, and 93 PhD programs. I identified 873 people as archaeologists on American faculties, comparable to the 875 reported from the 2007–08 *AnthroGuide*, and drew a 15% simple random sample (n = 131) for detailed study. (Statistical calculations cited below suggest that sample power justifies inferences.) Among them, 30 (23%) are in BA, 28 (21%) in MA, and 73 (56%) in PhD programs; none from satellite campuses fell in the sample. Proportions among degree programs are roughly comparable to larger samples drawn from different disciplines [13], although the archaeology sample has a slightly lower proportion of faculty in PhD programs.

## Merit: Definition and measurement

Faculty have many responsibilities, and most excel in only some of them. Yet this study starts from the proposition that "the reward system in science testifies that the research role is the most highly valued" ([33], 520). Of course other factors may contribute, but it assumes that

scholarly productivity chiefly determines initial hire and possible later recruitment to higher position. Often, productivity is estimated by number of publications and/or citations [2, 4, 7, 11–13, 32]. Measuring so complex and multifaceted an entity begs questions of quality vs. quantity, of weighting or not for scope of works (e.g. short essays versus books), of parsing credit among co-authors, of varying disciplinary publication rates.

Being confined to archaeology, the sample controls for at least broad disciplinary variation in publication rates [34]. I used Publish or Perish (PoP) [35] and Google Scholar (GS) for bibliographic indexing. GS indexes pre-prints, unpublished pdfs and other sources not necessarily peer-reviewed. This property, which may partly explain GS's high returns demands, in its own turn, extensive filtering [36, 37]. PoP facilitates filtering by, for instance, the ability to combine multiple entries for a single scholarly product, whether ms. versus published version, by misspelling of the title, or by simple duplication.

Journal articles may be the near-exclusive venue for scholarly output in many natural-science fields, but proceedings and books are common in other fields. Archaeology, like other social sciences [12, 32, 38], falls somewhere between the extremes defined by natural sciences, on the one hand, and humanities on the other. As a hybrid discipline, archaeological scholarship appears in dedicated archaeological and broader anthropology journals, in edited volumes and their chapters, and in books and monographs, which GS samples better than journal-based databases like Web of Science [39].

## Disambiguation

Once, GS searches could be confined to major disciplinary groups (e.g. social sciences). More recently, however, GS eliminated this option, forcing searches to survey its entire database. Therefore, searching archaeologists by name can return results for scholars in very different fields. Relatively few archaeologists have created GS profiles, which precluded PoP search using the GS Profile option.

Accordingly, this study confronted the challenge of name disambiguation. It posed little difficulty where surnames were uncommon. Yet the sample included a number of very common surnames, mostly of European origin. Following [35], I identified most scholars by their first initial and complete surname. In PoP searches, I took care to correctly spell archaeologists' names as they appeared in the sources consulted. Search for the scholarship of a young archaeologist who possesses a distinctive surname took only a few minutes. A comparable search for a senior scholar and/or one who possesses a common surname could require hours. The PoP search itself was a matter of moments, but disambiguation of similarly-named scholars, editing of the record for meeting papers and the like, pooling of multiple entries of the same work and, in some cases, detailed comparison of search results with the scholar's on-line curriculum vita, could take considerable time.

Search for each sampled archaeologist was conducted using PoP v5, in February and March 2017. Search interval began four years before award of PhD (to capture scholarship completed during the graduate career) and ended in 2017. For PoP's "Any of the keywords" the terms "archaeology" and "anthropology," were entered without quotation marks. Because virtually all academic archaeologists hold the PhD, dissertations were omitted from search results, although what often were published versions of them were included; book reviews also were omitted because they are not original research.

It would be fatuous to claim that all or nearly all scholarship by all sampled archaeologists was recorded by GS, because publication count is sensitive to sampling. But sampling should not affect major bibliographic indices that gauge impact rather than mere numbers of publications. Possible database effects upon measures like citations per co-author are unknown.

## Bibliometric merit measures

This study's focus is on scholarly influence as measured by citations, so it avoids altimetrics that may conflate productivity with mere attention. The University of California's MELVYL was used to measure publication count of American archaeologists as of academic year 1996–97 [30]. Since then, a "Cambrian explosion" (J. Bollen, cited in [40], 864) in bibliometric measurement has produced many indices (e.g., [35, 41]). From GS results, PoP computes nine bibliographic measures of scholarly productivity and impact, in particular:

- Hirsch's *h*, an integer denoting the number of publications that have been cited at least *h* times each. A popular modern index, *h* at once measures productivity (number of works) and impact (their aggregate reception).

- Egghe's *g*, which ranks sources in descending order the number of publications that together received $g^2$ or more citations.

- PoP hI_annual (*hIann*), the average annual increase in *h*, designed in part to control for disciplinary differences in publication norms, largely mooted in this study, and for the cumulative quality and therefore time-dependence of *h* (which can only rise and never decline, so long as citing sources remain accessible) that favors senior scholars. Thus, *hIann* permits more valid comparison between junior and senior scholars based upon *h* or related measures. Unlike *h*, *g* and other measures, *hIann* scales productivity by co-authorship.

PoP compiles other measures to control for co-authorship (individual *h*, hI, and hm) and cumulative effects (e.g. Age-weighted Citation Rate [AWCR], which adjusts for citation date). *h*, trivially, and others are "h-dependent," *g* among h-independent rank indicators [41]. Because disciplines vary in their number of practitioners and publication outlets, and in their norms for the frequency, amount and type of publication, bibliographic measures must be calibrated for comparison between fields. Subjects of this study all are archaeologists, which moots norming problems.

Because "it is always prudent to use several indicators to measure research performance" ([42], 5), analysis involved *h* for its intuitive clarity, *g* for its algorithmic independence of *h* and the weight assigned to highly influential papers whose citation counts exceed the *h*-core's minimum, and *hIann* for both its control over career length and its rewarding of sustained, not just episodic, production. [39] justifies *h* at length against sometimes harsh criticism and alternatives with which *h* correlates significantly (e.g. [43]). Jointly or separately, *h*, *g*, and *hIann* possess desirable qualities (e.g., quantifiability, robusticity, revisability, resistance to manipulation [44]). They are not, however, expressed with associated measures of uncertainty, or controlled for co-authorship; only *hIann* controls for career stage, i.e. is not cumulative. No single measure is comprehensive, no set of them ideal for all purposes. Exactitude seems questionable in scaling the scholarly merit of archaeologists. Mean and other measures for *hIann* was recorded to two decimal places, for integers *h* and *g* to one decimal place.

However calculated for whatever purpose, bibliographic measures often are correlated (e.g, [37, 40]). They are in this dataset; all pairwise correlations and rank-correlations among *h*, *g*, *hIann* and other PoP measures return positive coefficients, and p < .01. All PoP indices save *hIann* correlate at values near or above 0.90, *hIann* with all others at values ranging from 0.60–0.80. The slightly divergent results may owe to non-cumulative *hIann* measuring a rate that can vary in either direction over time, while cumulative indices never decline.

## Scholarly precocity

To measure precocity, a second PoP search for each sampled individual was confined to the period that began four years before the PhD year and ended two years after it, reasoning that

most graduate-study publication is likely to occur in the last four years before earning the doctorate. But hires also may be influenced by promise of imminent scholarly production (e.g., dissertations to be published as papers or books), so this interval was extended to two years past the PhD on the assumption that most scholarly promise would materialize by then.

Relevant measures of *h*, *g*, and *hIann* were recorded as prec_*h*, prec_*g* and prec_*hIann*, respectively. Unfortunately, PoP cannot compute these indices over just the search interval (A. Harzing, personal communication 6 March 2017). Therefore, cumulative measures like *h* and *g* pertain to the individual's entire career rather than the restricted search interval. Thus, if an archaeologist published three journal articles and one book chapter during the limited search interval that ended a decade or more ago, most or all probably were cited at least four times since then. If so, precocity measures approximate publication counts which, anyway, also should differ on average between young faculty hired into different program levels. Conversely, *hIann* values often were high for recent PhDs, whose productivity over the restricted search interval compared to senior colleagues was much nearer their career totals. As a result, relatively productive junior scholars apportion cumulative bibliographic measures over a shorter interval (and, therefore, smaller *hIann* denominator) that yields high values.

## Results

Analysis was conducted in SPSS™, statistical power calculations in powerandsamplesize.com. Number of archaeologists by department is low and right-skewed for BA and PhD programs, approximately normal for MA ones. Mean number of archaeologists per department rises by a factor roughly of two from BA through MA to PhD programs. Many archaeologists are widely scattered in ones and twos, mostly in BA programs, fewer in slightly larger MA ones. More archaeologists work in relatively few but large PhD programs.

Comparison between program levels or NRC ranks must consider the possible complicating factor of difference in career length or rank. A crude measure of career length is the difference between sample year, 2017, and year of PhD award ("professional age"). Obviously, this is an imperfect estimate of years-as-professor, but any error introduced is assumed to be minor. Pattern in on this measure is nearly significant (Kruskal-Wallis [K-W] $\chi^2$ = 4.9 p = .09; ANOVA F = 2.7 p = .07) only because mean longevity in MA programs (= 16.1 years) is considerably lower than in BA (= 20.8) and PhD (= 22.3) programs. There are no significant differences in the distribution of professorial rank (assistant, associate, "full") by degree program ($\chi^2$ = 2.80, p = .59; 0E<5) although the assistant rank is slightly overrepresented in MA programs (standardized residual = 1.2), or in pooled BA/MA versus PhD program ($\chi^2$ = 0.64, p = .73; 0E<5). In sum, there are no clear seniority differences by program level; differences between them in scholarly productivity owe to other factors.

### Precocity measures

Universalism implies that initial hire is based mostly on precocity, the number and quality of publications already published or under review at hire. Early-career productivity may be disproportionately valued, creating "an institutionalized bias in favor of precocity" ([10], 613). It can influence initial hire if more prestigious institutions preferentially hire more productive new PhDs [5]. Yet some sources [2, 11] found no correlation between precocity and career-long measures nor any between predoctoral productivity and institutional prestige of the initial hire.

Precocity measures correlate strongly and significantly with one another (e.g., r = .95 for prec_*h* and prec_*g*). They also correlate significantly with their equivalent career measures, at much lower values (e.g., r = .51 for prec_*h* and *h*; r = .47 for prec_*g* and *g*). Precocity gauges

**Table 1. Precocity and career bibliographic measures by program level.**

| | prec_h | | prec_g | | prec_hIann | |
|---|---|---|---|---|---|---|
| | mean | s.d. | mean | s.d. | mean | s.d. |
| BA | 1.8 | 1.7 | 2.5 | 2.3 | 0.11 | 0.16 |
| MA | 3.1 | 3.0 | 4.1 | 4.0 | 0.20 | 0.18 |
| PhD | 3.1 | 2.5 | 4.7 | 4.8 | 0.16 | 0.14 |
| BA+MA | 2.4 | 2.5 | 3.3 | 3.4 | 0.15 | 0.18 |
| | prec_articles | | prec_chapters | | prec_books | |
| | mean | s.d. | mean | s.d. | mean | s.d. |
| BA | 2.2 | 2.2 | 0.5 | 1 | 0.07 | 0.3 |
| MA | 3.3 | 3.3 | 1.07 | 1.4 | 0.18 | 0.4 |
| PhD | 3.5 | 4.0 | 0.93 | 1.3 | 0.23 | 0.6 |
| | h | | g | | hIann | |
| | mean | s.d. | mean | s.d. | mean | s.d. |
| BA | 6.3 | 6.4 | 11.7 | 11.7 | 0.27 | 0.19 |
| MA | 7.3 | 6.0 | 13.2 | 12.4 | 0.33 | 0.20 |
| PhD | 13.1 | 9.5 | 25.5 | 19.3 | 0.43 | 0.19 |
| BA+MA | 6.9 | 6.1 | 12.6 | 12.0 | 0.30 | 0.19 |

scholarly records near point-of-hire, so tests the first universalist corollary: differential initial placement by merit. prec_$h$ and prec_$g$ pattern consistently by degree program, although mean differences are modest; prec_$hIann$ does not pattern consistently; the MA-program mean exceeds the PhD mean (Table 1, Fig 1) (prec_$h$ K-W $\chi^2$ = 8.7 p = .01; prec_$g$ K-W $\chi^2$ = 7.4 p = .03; prec_$hIann$ K-W $\chi^2$ = 6.2 p = .05; parametric F also differs significantly by degree program, where pairwise least-significant-difference [LSD] p≤.05 in prec_$h$ and prec_$g$ only between BA and PhD programs, and in prec_$hIann$ only between BA and MA programs). Statistical power relates to Type II errors; all significant variable differences exceeded the acceptable power level of 0.80. Differences in precocity measures lie mostly between archaeologists in BA versus higher programs. Results fail to attain significance when comparing archaeologists in MA and PhD programs (prec_$h$ Mann-Whitney U = 918.5 p = .43; prec_$g$ U = 915.0 p = .41; prec_$hIann$ U = 905.5 p = .35).

The (presumably higher) productivity of hires from higher-ranked institutions may explain via universalism the hiring patterns otherwise attributed to particularist networks [7, 9], if slight differences in productivity distributions between candidates from higher- and lower-ranked institutions are proportionally high at the distributions' upper tails [45]. This

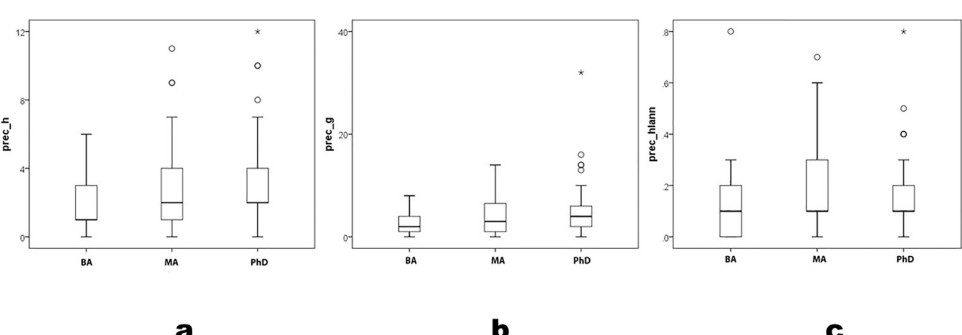

**Fig 1. prec_$h$, prec_$g$ and prec_$hIann$ by degree program.**

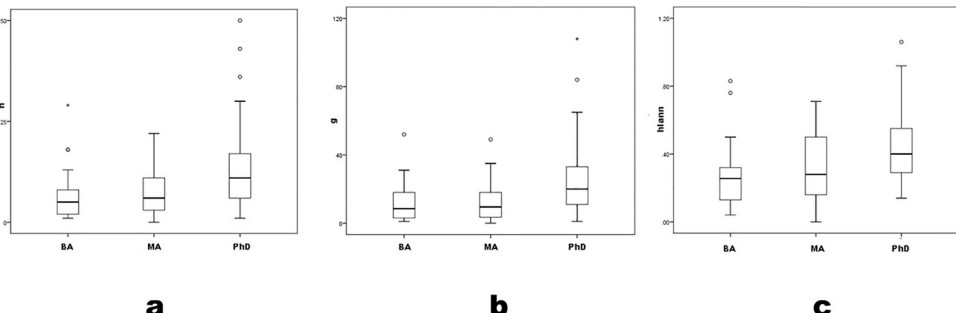

**Fig 2. *h*, *g* and *hIann* by degree program.**

theoretical argument is not yet demonstrated in evidence. Besides journal-impact factors as sole merit measure, it assumes complete information about candidates' comparative records, and competition for jobs based solely upon those journal impact factors. Yet in "Anglo-Saxon countries. . .the curriculum vitae, interview process, and letters of recommendation 'count' more than the bibliometric indicators" ([46], 3). However valid assumptions about distributions' tails may be elsewhere, they are not at face value applicable to archaeology, where hiring practices resemble the "Anglo-Saxon" norm, and other factors (e.g., area and theoretical emphases, lack of disciplinary consensus as well as network effects) influence the hiring process.

Precocity bibliometric indices do not sort archaeologists clearly by degree program (see [2, 14] for similar conclusions elsewhere). Precocity measures distinguish the BA from higher levels. Archaeologists in MA and PhD programs are statistically indistinguishable. Near hire, little distinguishes archaeologists by degree programs, and effectively nothing distinguishes those in MA and PhD programs. Universalism's first corollary—placement determined by merit—is unevenly supported in these data.

## Career measures

All bibliographic indices differ significantly across degree programs (Table 1, Fig 2) (*h* K-W $\chi^2$ = 21.5 p < .01; *g* K-W $\chi^2$ = 22.7 p < .01; *hIann* K-W $\chi^2$ = 16.8 p < .01; parametric F also differs significantly by degree program, where no pairwise LSD≤.05 between BA and MA programs and all LSD≤.05 between PhD and both lower programs).

Thus, archaeologists in BA and MA programs form a single subsample for comparison to archaeologists in PhD programs. So grouped, productivity still patterns by degree program (Fig 3), significantly (*h* U = 1133.0 p < .01; *g* U = 1095.0 p < .01; *hIann* U = 1276.0 p < .01;

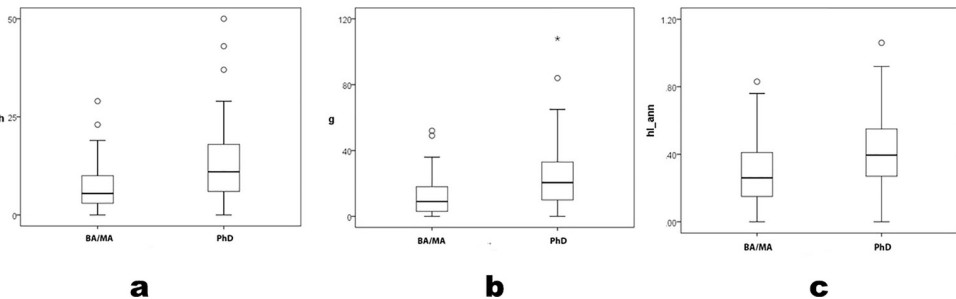

**Fig 3. *h*, *g* and *hIann* by grouped BA+MA versus PhD program.**

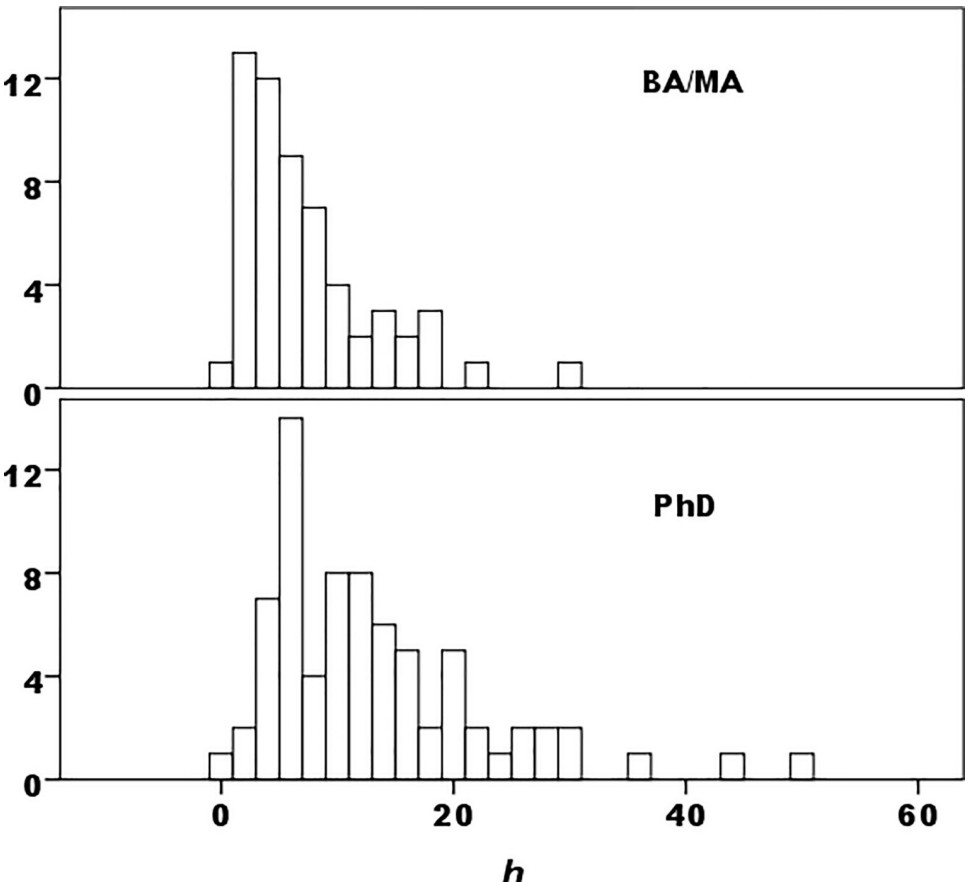

**Fig 4. Distribution of *h* by combined BA+MA versus PhD program.**

parametric Student's t yields similar conclusions, and power levels = 1.00 for all statistical differences). Nevertheless, measures' ranges do not sort clearly by program level, but overlap considerably; many BA- and MA-program archaeologists are as productive as most PhD-program ones. Universalism's second corollary—merit-sorting by hierarchical level—is supported statistically but with exceptions difficult to accommodate to the ideal.

In career measures, pairwise comparison distinguishes PhD-program archaeologists as more productive on average, despite the absence of clear differences in corresponding precocity measures between PhD and MA programs. If productivity were primarily an individual quality, then the absence of difference in precocity measures by degree program should persist in career ones. It does not.

**Distribution effects.** Productivity skewness is typical of many scholarly fields (e.g., [2, 4, 6, 11]). In this sample, measures like *h* are right-skewed, particularly at the PhD level (e.g. Fig 4). Yet with a mode lying at 1 *h* also is left-skewed in the BA/MA subset, perhaps more difficult to perceive in the constrained range of low variation in *h*. Skewing obviously justifies the use of nonparametric statistics. To gauge effects of skews left and right, a trimmed sample omitted the top quartile in PhD programs (19 of 74) and BA/MA-program archaeologists whose $h<2$, clearly those not committed to scholarship (13 of 58, 22.4%). To prevent the few PhD archaeologists whose $h<2$ (n = 3) to affect results, they too were omitted. Overall, this treatment omits one subset's top quartile and its extreme lower tail, the other's approximate bottom quartile (35 of 132, 26.5%).

**Table 2. Trimmed-sample productivity means, overall and by specified cohorts of professional age (years since PhD).**

| | | | ALL | | | COHORTS | | | | |
|---|---|---|---|---|---|---|---|---|---|---|
| Prog. | | : | $h$ | | : | $g$ | | : | $hIann$ | |
| Level | n | : | mean | s.d. | : | mean | s.d. | : | mean | s.d. |
| BA | 23 | : | 8.0 | 6.4 | : | 14.8 | 11.6 | : | 0.31 | 0.19 |
| MA | 22 | : | 9.0 | 5.4 | : | 16.5 | 12.0 | : | 0.39 | 0.18 |
| PhD | 52 | : | 8.9 | 4.0 | : | 17.1 | 8.1 | : | 0.33 | 0.12 |
| BA/MA | 45 | : | 8.5 | 5.8 | : | 15.6 | 11.7 | : | 0.35 | 0.19 |
| PhD | 52 | : | 8.9 | 4.0 | : | 17.1 | 8.1 | : | 0.39 | 0.12 |
| | | | PhD | YEAR | | <2002 | | | | |
| Prog. | | : | $h$ | | : | $g$ | | : | $hIann$ | |
| Level | n | : | mean | s.d. | : | mean | s.d. | : | mean | s.d. |
| BA | 15 | : | 9.4 | 7.5 | : | 18.4 | 13.0 | : | 0.28 | 0.17 |
| MA | 11 | : | 12.1 | 4.8 | : | 22.8 | 12.2 | : | 0.40 | 0.16 |
| PhD | 26 | : | 10.4 | 3.4 | : | 20.4 | 7.5 | : | 0.34 | 0.144 |
| BA/MA | 26 | : | 10.5 | 6.5 | : | 20.3 | 12.6 | : | 0.33 | 0.17 |
| PhD | 26 | : | 10.4 | 3.4 | : | 20.4 | 7.5 | : | 0.34 | 0.14 |
| | | | PhD | YEAR | | 2007–2017 | | | | |
| Prog. | | : | $h$ | | : | $g$ | | : | $hIann$ | |
| Level | n | : | mean | s.d. | : | mean | s.d. | : | mean | s.d. |
| BA/MA | 7 | : | 5.0 | 1.8 | : | 7.3 | 2.8 | : | 0.46 | 0.24 |
| PhD | 11 | : | 6.3 | 3.6 | : | 12.3 | 9.1 | : | 0.46 | 0.19 |
| | | | PhD | YEAR | | 1997–2006 | | | | |
| Prog. | | : | $h$ | | : | $g$ | | : | $hIann$ | |
| Level | n | : | mean | s.d. | : | mean | s.d. | : | mean | s.d. |
| BA/MA | 20 | : | 6.5 | 3.9 | : | 12.0 | 8.5 | : | 0.33 | 0.17 |
| PhD | 21 | : | 8.4 | 3.9 | : | 15.3 | 5.7 | : | 0.42 | 0.16 |
| | | | PhD | YEAR | | ≤1996 | | | | |
| Prog. | | : | $h$ | | : | $g$ | | : | $hIann$ | |
| Level | n | : | mean | s.d. | : | mean | s.d. | : | mean | s.d. |
| BA/MA | 18 | : | 12.1 | 6.9 | : | 22.9 | 13.0 | : | 0.32 | 0.17 |
| PhD | 19 | : | 10.5 | 3.4 | : | 21.4 | 8.0 | : | 0.31 | 0.13 |

Here, differences are insignificant (Table 2) (across all three program levels $h$ K-W $\chi^2$ = 2.63 p = .269; $g$ K-W $\chi^2$ = 2.61 p = .271; $hIann$ K-W $\chi^2$ = 1.52 p = .221; for combined BA/MA versus PhD $h$ U = 1005.0 p = .231; $g$ U = 963.5 p = .135; $hIann$ U = 1194.5 p = .766; again, parametric ANOVA F and Student's t yields similar conclusions). For brevity, Fig 5 shows only paired BA/MA versus PhD subsets. Finally, comparing the $h$ and $g$ upper quartile of BA/MA affiliates (n = 15) to all PhD affiliates (n = 74), the former's $h$, $g$, and $hIann$ means all exceed PhD means, insignificantly ($h$ U = 385.0 p = .062; $g$ U = 393.0 p = .076; $hIann$ U = 460.5 p = .298; again, Student's t yields similar conclusions).

There are significant statistical differences between BA/MA and PhD subsets. The trimmed sample does not elide differences, but locates them more precisely than a crude distinction among archaeologists by program level can. As a group, archaeologists in PhD programs clearly outperform others over time, from causes either universal or particular or some combination of both. But when their top quartile and the (approximate) low-achieving bottom quartile of BA/MA archaeologists are excluded, productivity differences fade to insignificance. For the great majority of academic archaeologists—the trimmed sample, about three-quarters of each subset—difference in placement by program level does not correspond to difference in

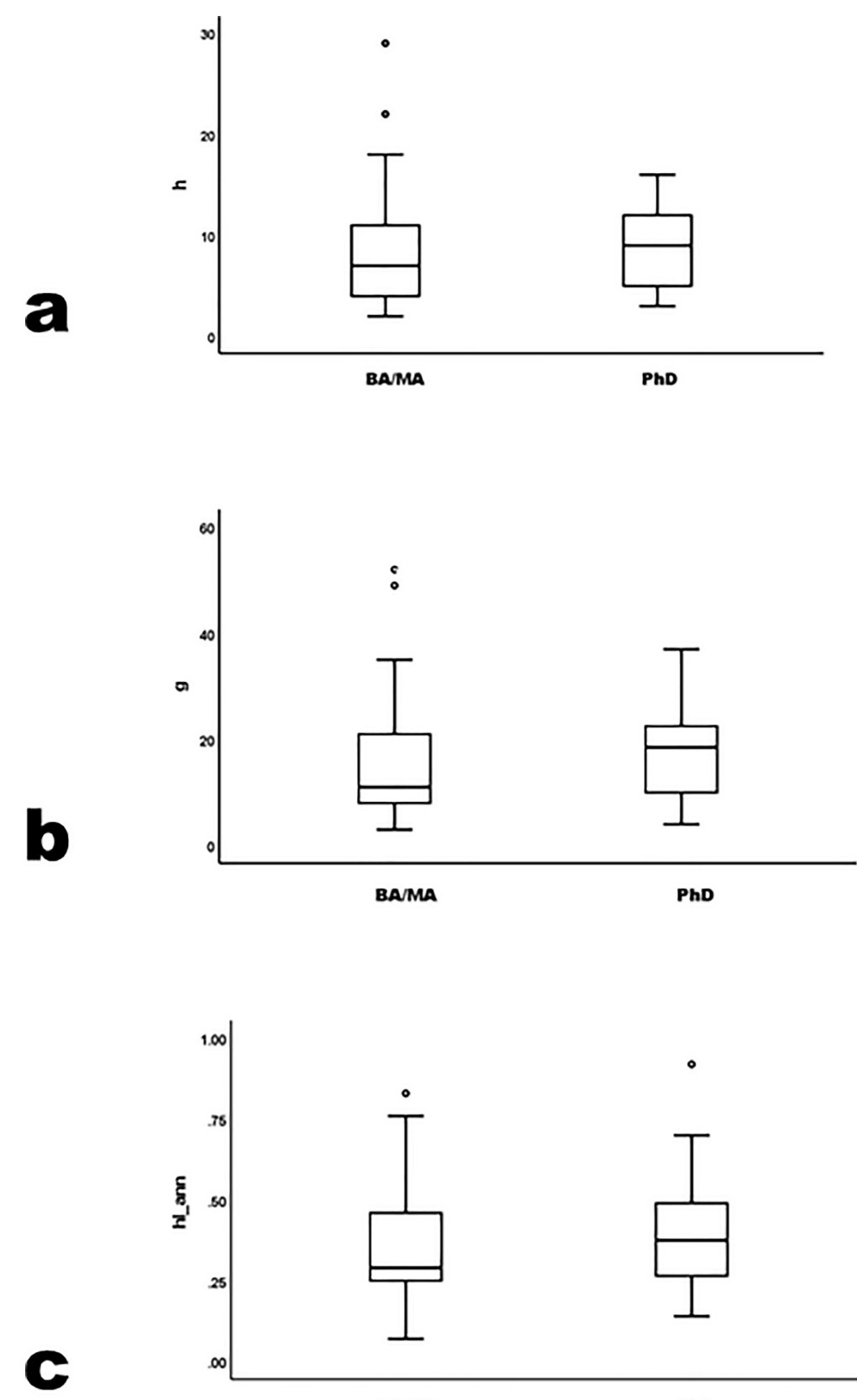

**Fig 5. Trimmed-sample *h*, *g* and *hIann* by combined BA+MA versus PhD program.**

**Table 3. Precocity and career bibliographic measures (mean, standard deviation [s.d.]) by program level for PhD year < 2002. [1].**

| | prec_h | | prec_g | | prec_hIann | |
|---|---|---|---|---|---|---|
| | mean | s.d. | mean | s.d. | Mean | s.d. |
| BA | 1.7 | 1.6 | 2.1 | 2 | 0.05 | 0.07 |
| MA | 3.4 | 3.4 | 4.6 | 4.8 | 0.12 | 0.09 |
| PhD | 2.9 | 2.3 | 3.9 | 3.3 | 0.1 | 0.08 |
| BA+MA | 2.4 | 2.6 | 3.2 | 3.6 | 0.08 | 0.08 |
| | prec_articles | | prec_chapters | | prec_books | |
| | mean | s.d. | mean | s.d. | mean | s.d. |
| BA | 1.9 | 1.7 | 0.2 | 0.4 | 0.12 | 0.33 |
| MA | 3.3 | 3.4 | 1.5 | 1.8 | 0.23 | 0.44 |
| PhD | 2.8 | 2.9 | 0.8 | 1.1 | 0.23 | 0.57 |
| | h | | g | | hIann | |
| | mean | s.d. | mean | s.d. | mean | s.d. |
| BA | 8.4 | 7.6 | 16.4 | 13.4 | 0.25 | 0.18 |
| MA | 10.4 | 6.3 | 19.5 | 13.8 | 0.34 | 0.2 |
| PhD | 16.7 | 9.9 | 33 | 20.4 | 0.43 | 0.18 |
| BA+MA | 9.3 | 7 | 17.8 | 13.4 | 0.29 | 0.19 |

[1]BA n = 17, MA n = 13, PhD n = 44.

scholarly productivity. Nor do productivity differences extend to the upper range of BA/MA affiliates, whose top $h$ quartile exceeds, insignificantly, PhD affiliates in all mean values.

**Seniority effects.** As above, faculty rank is distributed similarly among program levels; there are no significant differences in seniority or inferred longevity by program. Nevertheless, $h$ and $g$ are cumulative career measures that justify subdivision of the dataset. This was accomplished first, by coarsely defining a subsample of archaeologists whose PhD year was earlier than 2002. This subsample is of professional age of 15+ years.

Precocity measures calculated in the senior subsample do not differ by degree program (prec_$h$ K-W $\chi^2$ = 4.9 p = .09; $g$ K-W $\chi^2$ = 4.3 p = .12; $hIann$ K-W $\chi^2$ = 5.5 p = .07). In career measures, differences between PhD and lower programs persist in this subsample (Table 3; Fig 6) ($h$ K-W $\chi^2$ = 14.2 p < .01; $g$ K-W $\chi^2$ = 13.7 p < .01; $hIann$ K-W $\chi^2$ = 13.3 p < .01; again parametric ANOVA patterns similarly and most pairwise LSD p≤.05 between PhD and lower programs although $hIann$ LSD p = .17 between MA and PhD programs). Again, therefore, BA and MA groups were combined for comparison to the PhD subsample. Again, the latter significantly exceed the former (for all three measures U≥330 and p < .01, parametric Student's t gives similar results, and statistical power>0.80). Again, however, the trimmed subsample yields different results. In career measures, differences by program level fade (Fig 7) (across all three program levels $h$ K-W $\chi^2$ = 3.19 p = .203; $g$ K-W $\chi^2$ = 1.64 p = .441; $hIann$ K-W $\chi^2$ = 3.91 p = .141; for combined BA/MA versus PhD $h$ U = 310.5 p = .613; $g$ U = 307.5 p = .576; $hIann$ U = 324.0 p = .798; again, parametric ANOVA F and Student's t yields similar conclusions). Also, MA-program means exceeds PhD means, as does $h$ for the BA/MA group.

The sample can be subdivided even more finely by defining cohorts whose professional age ranges from 0 to 10, 11–20 and 20+ (i.e., PhD years falling between 2007–2017, 1997–2006, before 1996) (see [32], 870–873 for similar treatment of academic sociologists). Again combining BA and MA groups, mean values of bibliometric measures pattern as expected in two ways (Table 4) (Student's t gives similar results to U; all statistical power calculations>0.80). First, usually the PhD-program mean value is significantly higher. Second, the difference in mean $h$

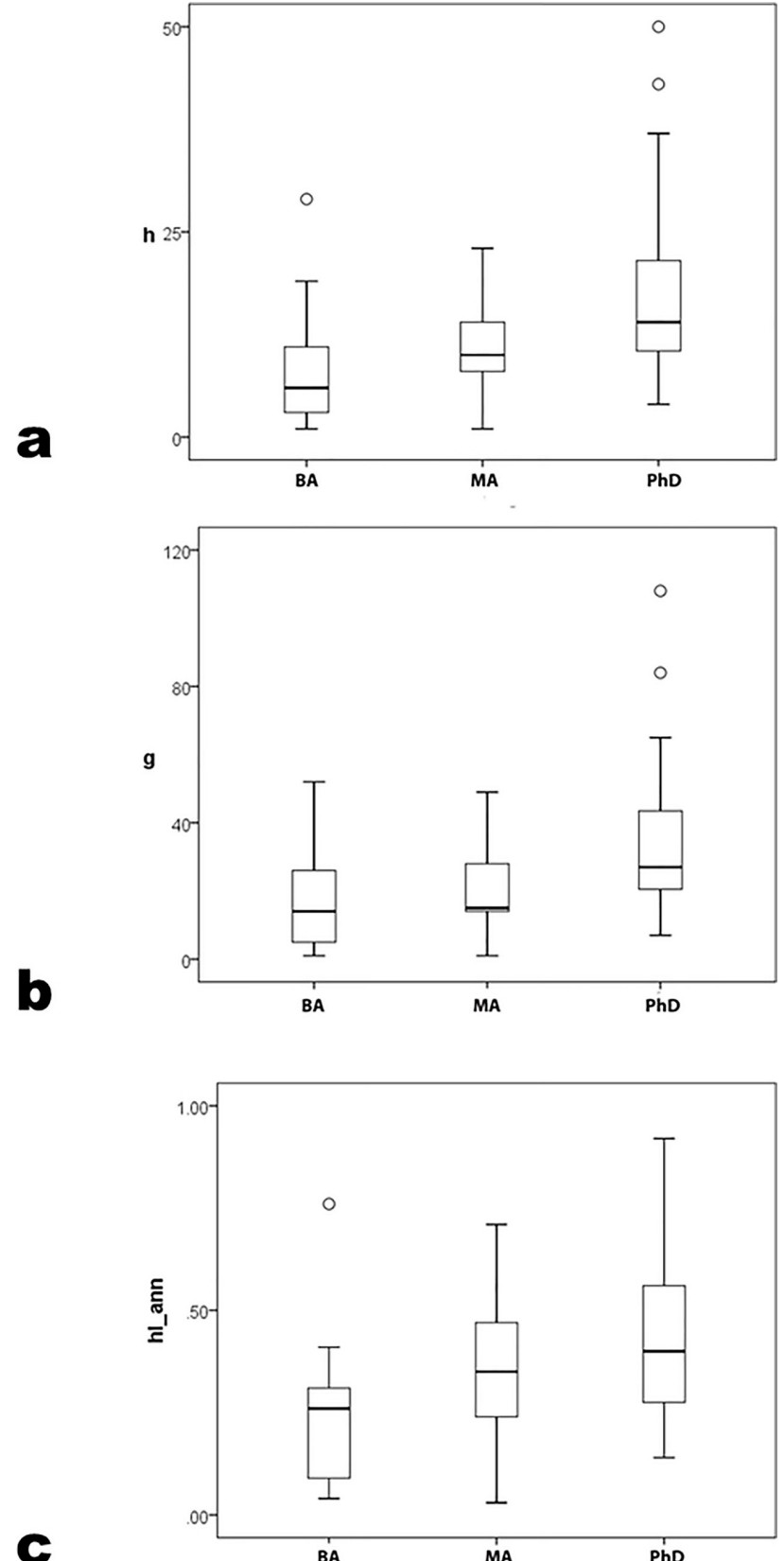

**Fig 6. *h*, *g* and *hIann* by degree program for archaeologists who earned the PhD before 2002.**

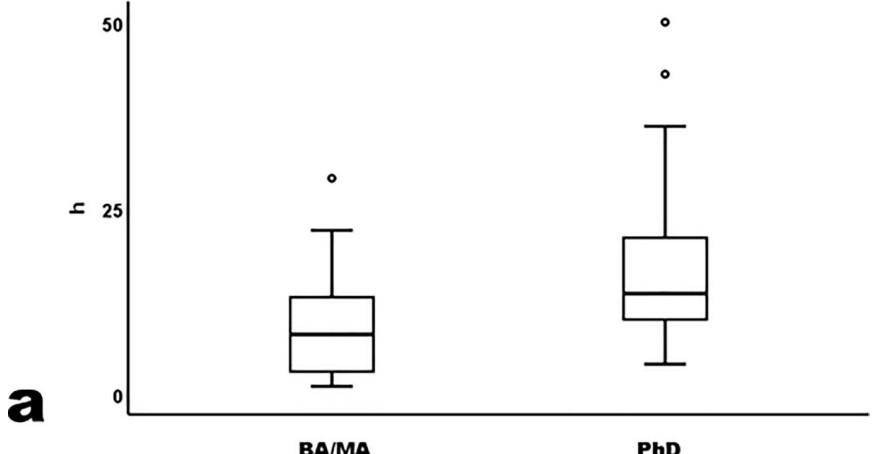

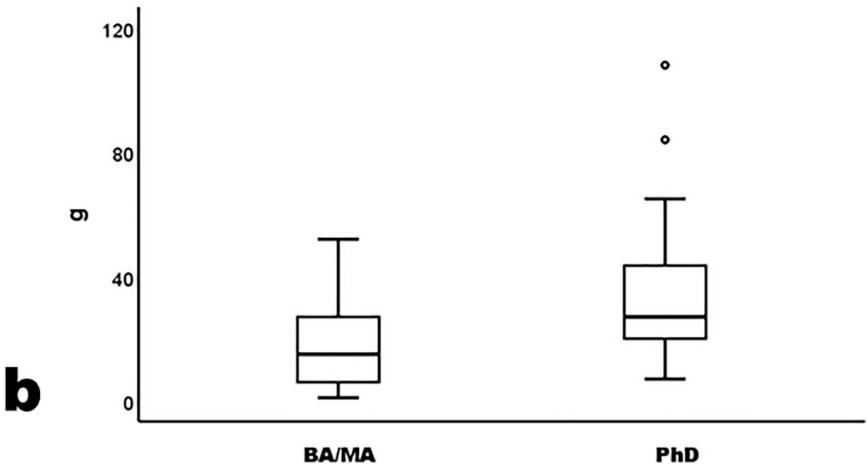

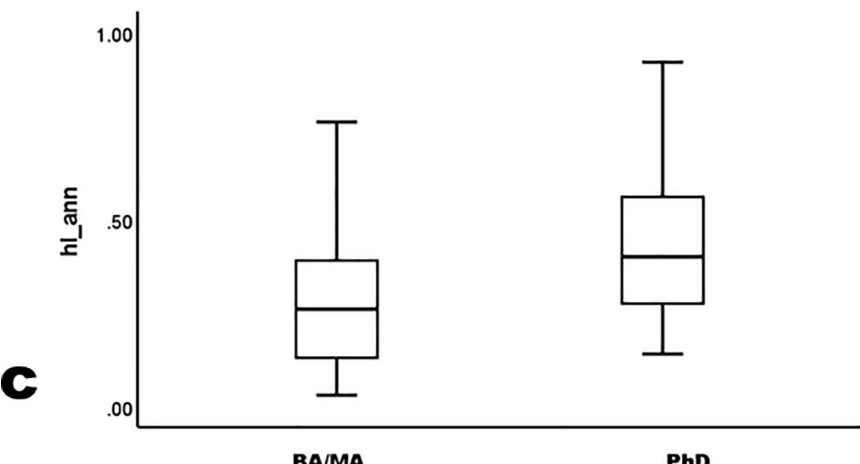

**Fig 7. Trimmed sample *h*, *g* and *hIann* by grouped degree program for archaeologists who earned the PhD before 2002.**

and *g* values increases steadily from youngest to oldest cohort, as in *hIann* from youngest to second cohort, broadly consistent with ([15], Table 1). The trimmed sample patterns differently, although low counts confine comparison to BA/MA versus PhD. Differences remain significant for the middle cohort (PhD year 1997–2006) but are not significant for others (for PhD year 2007–2017 *h* U = 31.5 p = .536; *g* U = 24.0 p = .211; *hIann* U = 36.0 p = .860; for PhD year 1997–2006 *h* U = 147.0 p = .098; *g* U = 133.5 p = .045; *hIann* U = 146.5 p = .097; for PhD year<2006 *h* U = 48.0 p = .702; *g* U = 53.5 p = .972; *hIann* U = 48.5 p = .702; again, parametric Student's t yields similar conclusions), the BA/MA *hIann* mean is slightly higher in the 2007–2017 cohort, and all BA/MA means exceed PhD ones in the oldest cohort.

Career measures *h* and *g* cumulate over time, so may conflate longevity and productivity. *hIann* is a rate that does not cumulate, so should be independent of career length yet does tend to rise between first and second career decades. In seniority cohorts, precocity and career measures pattern as they do in the entire sample. In older cohorts, archaeologists in PhD programs increase their advantage over BA/MA colleagues. Productivity differences are not a function of longevity (although their magnitudes are), or cumulative measures nor, as above, of academic rank. The only salient factor to explain productivity differences is degree program. That difference does not exist before initial hire but correlates with later productivity. The trimmed sample, however, mostly fails to distinguish archaeologists' productivity by program level, locating differences chiefly at the tails of their respective distributions. Accordingly, for about three-quarters of the sample there are no significant productivity differences by program level, suggesting both great productivity overlap between program levels and the absence of consistent universal factors to explain affiliation. Finally, the top *h* quartile of BA/MA affiliates statistically is indistinguishable from PhD affiliates.

**NRC ranking among PhD programs.**   Data also can be interrogated for the relationship between merit and placement among PhD-program archaeologists, accomplished using NRC program rankings. Reported results are from analysis of individual default rankings compiled by NRC [47].

**Table 4. Mean career index values by degree program, and difference between PhD and BA/MA means, by cohort. [1].**

| PhD YR | | | | | | n | h | g | hI_ann |
|---|---|---|---|---|---|---|---|---|---|
| 2007–2017 | BA/MA | | | | | 15 | 2.8 | 4.1 | 0.31 |
| 2007–2017 | PhD | | | | | 13 | 5.5 | 10.6 | 0.42 |
| | | | | | | | U = 52.5 p = .04 | U = 44.5 p = .01 | U = 58.0 p = .07 |
| | | | PhDμ minus BA/MAμ | | | | 2.7 | 6.5 | 0.11 |
| 1997–2006 | BA/MA | | | | | 22 | 6 | 11 | 0.31 |
| 1997–2006 | PhD | | | | | 23 | 10.6 | 18.8 | 0.46 |
| | | | | | | | U = 147 p = .01 | U = 133.5 p < .01 | U = 146.5 p = .02 |
| | | | PhDμ minus BA/MAμ | | | | 4.5 | 7.8 | 0.15 |
| <1997 | BA/MA | | | | | 21 | 10.5 | 19.9 | 0.28 |
| <1997 | PhD | | | | | 36 | 17.5 | 35.1 | 0.42 |
| | | | | | | | U = 208 p < .01 | U = 196.5 p < .01 | U = 227 p < .01 |
| | | | PhDμ minus BA/MAμ | | | | 7 | 15.3 | 0.14 |

[1]Mann-Whitney U and attained-significance reported.

**Table 5. Precocity and career measures (mean, standard deviation [s.d.]) by NRC rank upper and lower halves, and descending quartiles.**

| | prec_h | | prec_g | | prec_hIann | |
|---|---|---|---|---|---|---|
| | mean | s.d. | mean | s.d. | mean | s.d. |
| Upper[1] | 3.6 | 2.3 | 4.9 | 3.1 | 0.17 | 0.12 |
| Lower[1] | 3 | 2.8 | 4.9 | 6.7 | 0.17 | 0.18 |
| | h | | g | | hIann | |
| | mean | s.d. | mean | s.d. | mean | s.d. |
| Upper | 16.8 | 11.7 | 32.2 | 23.5 | 0.49 | 0.21 |
| Lower | 10.5 | 6.8 | 20.1 | 12.7 | 0.42 | 0.17 |
| | prec_h | | prec_g | | prec_hIann | |
| | mean | s.d. | mean | s.d. | mean | s.d. |
| 1st[2] | 4 | 2.7 | 5.5 | 3.5 | 0.18 | 0.12 |
| 2nd[2] | 3.2 | 1.8 | 4.2 | 2.8 | 0.15 | 0.12 |
| 3rd[2] | 2.5 | 2.3 | 3.7 | 4.1 | 0.14 | 0.13 |
| 4th[2] | 3.3 | 3.1 | 5.7 | 7.9 | 0.19 | 0.17 |
| | h | | g | | hIann | |
| | mean | s.d. | mean | s.d. | mean | s.d. |
| 1st | 18.9 | 11.7 | 37.5 | 23.8 | 0.53 | 0.24 |
| 2nd | 14.1 | 11.4 | 25.8 | 22.3 | 0.43 | 0.16 |
| 3rd | 11.6 | 9.4 | 22.7 | 17.3 | 0.36 | 0.12 |
| 4th | 9.8 | 4.8 | 18.4 | 9.1 | 0.45 | 0.2 |

[1]Upper n = 29, Lower n = 29.

[2]1st quartile n = 16, 2nd n = 13, 3rd n = 11, 4th n = 18.

Distinguishing archaeologists in the top and bottom halves of the NRC rankings yields insignificant results for precocity measures in nonparametric (prec_*h* U = 313.0 p = .09; prec_*g* U = 299.5 p = .06; prec_*hIann* U = 376.5 p = .47) (Table 5; Fig 8) and parametric analysis (prec_*h* t = 0.91 p = .37, prec_*g* t = 0.00 p = 1.00, prec_*hIann* t = 0.09 p = .93). Difference, particularly in prec_*g*, owes to a positive outlier in the bottom-half group that U and t weight differently. There is at best equivocal support for distinguishing between upper and lower NRC halves in prec_*h* and prec_*g*; prec_*hIann* does not differ by NRC halves.

However, most career-cumulative measures differ (Fig 9) (*h* U = 277 p = .03; *g* U = 281.0 p = .03; but *hIann* U = 347.5 p = .26; Student's t gave comparable results; power level for *h* and *g*>0.80 but only = 0.63 for *hIann*). As in comparison by degree programs, subdivision of PhD programs by NRC rank reveals no difference in precocity measures but significant ones in career *h* and *g*. ([[32], Tables 4 and 6]] results for publication counts by similar NRC tiers for sociology is a useful comparison, but did not test for differences in mean figures by tier.) Similarly, among PhD programs sorted by quartiles of rank, precocity measures do not differ significantly (Table 5; Fig 10) (prec_*h* K-W $\chi^2$ = 3.8 p = .29; prec_*g* K-W $\chi^2$ = 4.6 p = .21; prec_*hIann* K-W $\chi^2$ = 1.1 p = .79; parametric ANOVA gave similar results, where no pairwise LSD p≤.05). The fourth (lowest) quartile's means on all precocity variables exceed the third's value. Results are not significant in *h* (K-W $\chi^2$ = 7.5 p = .06) but are in *g* (K-W $\chi^2$ = 9.2 p = .03); *hIann* did not pattern consistently by quartile (K-W $\chi^2$ = 4.3 p = .23) (Fig 11). Parametric ANOVA patterned similarly; in pairwise comparisons, LSD p≤.05 between the fourth and first groups in *h* and *g* and between third and first in *hIann*.

As in comparison by degree program, there are no significant differences in precocity measures by halves or quartiles of NRC rank. Again, faculty of equal original productivity are

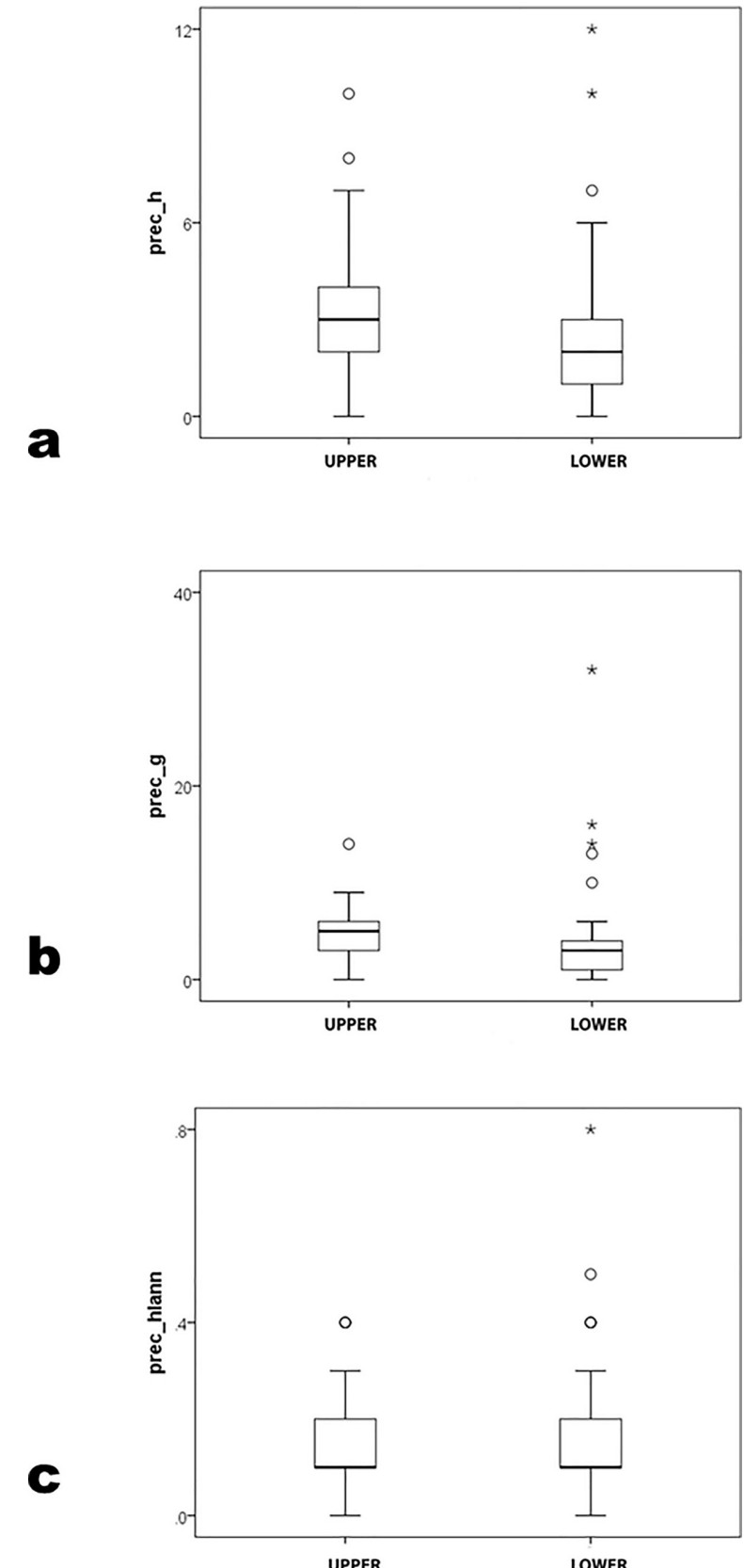

**Fig 8. prec_*h*, prec_*g* and prec_*hIann* by NRC-rank top and bottom halves.**

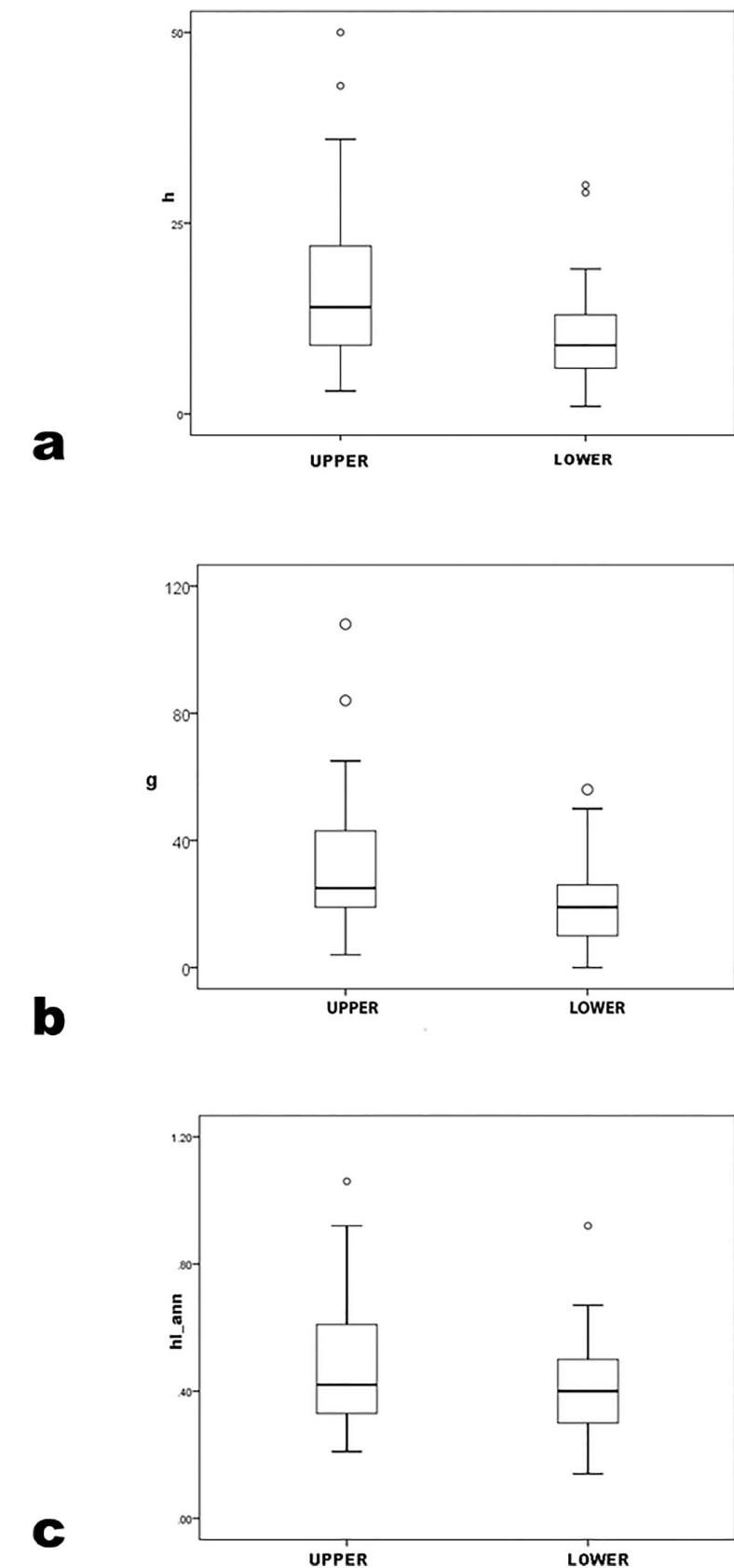

**Fig 9. _h_, _g_ and _hIann_ by NRC-rank top and bottom halves.**

**Table 6. Observed, meritocratic, and random allocation to program level (BA, MA, NRC-bottom-half PhD, NRC-top-half PhD) by quartiles ("qtl") of *h* and *hIann*.[1].**

| OBSERVED | program | level | | | | OBSERVED | program | level | | | |
|---|---|---|---|---|---|---|---|---|---|---|---|
| MERIT | BA | MA | PhDbot | PhDtop | Σ | MERIT | BA | MA | PhDbot | PhDtop | Σ |
| h qtl1 | 8 | 3 | 4 | 0 | 15 | hIann_qtl1 | 6 | 2 | 5 | 1 | 14 |
| h qtl2 | 2 | 2 | 8 | 3 | 15 | hIann_qtl2 | 5 | 3 | 4 | 4 | 16 |
| h qtl3 | 3 | 3 | 6 | 5 | 17 | hIann_qtl3 | 2 | 2 | 8 | 3 | 15 |
| h qt4 | 1 | 1 | 4 | 8 | 14 | hIann_qtl4 | 1 | 2 | 5 | 8 | 16 |
| Σ | 14 | 9 | 22 | 16 | 61 | Σ | 14 | 9 | 22 | 16 | 61 |
| MERITOCRATIC | | | | | | MERITOCRATIC | | | | | |
| h qtl1 | 14 | 1 | 0 | 0 | 15 | hIann_qtl1 | 14 | 0 | 0 | 0 | 14 |
| h qtl2 | 0 | 8 | 7 | 0 | 15 | hIann_qtl2 | 0 | 9 | 7 | 0 | 16 |
| h qtl3 | 0 | 0 | 15 | 2 | 17 | hIann_qtl3 | 0 | 0 | 15 | 0 | 15 |
| h qt4 | 0 | 0 | 0 | 14 | 14 | hIann_qtl4 | 0 | 0 | 0 | 16 | 16 |
| Σ | 14 | 9 | 22 | 16 | 61 | Σ | 14 | 9 | 22 | 16 | 61 |
| RANDOM | | | | | | RANDOM | | | | | |
| h qtl1 | 3.44 | 2.21 | 5.41 | 3.93 | 15 | hIann_qtl1 | 3.21 | 2.07 | 5.05 | 3.67 | 14 |
| h qtl2 | 3.44 | 2.21 | 5.41 | 3.93 | 15 | hIann_qtl2 | 3.67 | 2.36 | 5.77 | 4.20 | 16 |
| h qtl3 | 3.90 | 2.51 | 6.13 | 4.46 | 17 | hIann_qtl3 | 3.44 | 2.21 | 5.41 | 3.93 | 15 |
| h qt4 | 3.21 | 2.07 | 5.05 | 3.68 | 14 | hIann_qtl4 | 3.67 | 2.36 | 5.77 | 4.20 | 16 |
| Σ | 14 | 9 | 22 | 16 | 61 | Σ | 14 | 9 | 22 | 16 | 61 |

[1]In counts converted to proportions per 1,000, for *h*, Observed vs. Meritocratic dissimilarity index = .443, Observed vs. Random dissimilarity index = .218. For *hIann*, Observed vs. Meritocratic dissimilarity index = .526, Observed vs. Random dissimilarity index = .182.

recruited to programs of differing rank. Some differences emerge in career measures, often but not always by ascending NRC rank groups. Again, therefore, differences arise after hire.

## Mobility effects

A possible explanation for higher career-merit measures in PhD programs is mobility–recruitment to higher position after initial hire. On evidence here, junior scholars may be difficult to distinguish, but perhaps more productive scholars at lower ranks eventually are recruited to higher ones. Yet for a generation or more mobility has been notoriously constrained by academic archaeology's poor labor market, described euphemistically as "small, highly competitive" ([48], 291), perhaps more accurately as "bleak" ([9], 4) if merit is assumed to offer reasonable prospects for advancement) and the opportunity costs that attend tenure commitments.

Almost two-thirds of the sample—87 archaeologists—did not appear in a comparable 1997 sample [30]. Estimating from the current sample, about 580 of 873 archaeologists in the dataset were hired since 1997. (This estimate assumes complete sampling in 1997 and 2017, and that hiring was only of US PhDs. Neither assumption is entirely valid, but error they introduce is apt to be slight compared to the size of resulting estimates.) In the abstract, the number of new hires is substantial, but figures must be calibrated against the production rate of archaeology PhDs. No relevant data over the 1997–2017 span were found, but between 1971 and 2002 US anthropology departments produced an average of about 90 archaeology PhDs per year, omitting academic years 1989–1992 for which data were unavailable [49]. Between 1997 and 2002, they produced on average 116 PhDs per year. There is no reason to suspect that US programs have produced archaeology PhDs at lower rates since 2002. Nevertheless, roughly splitting the

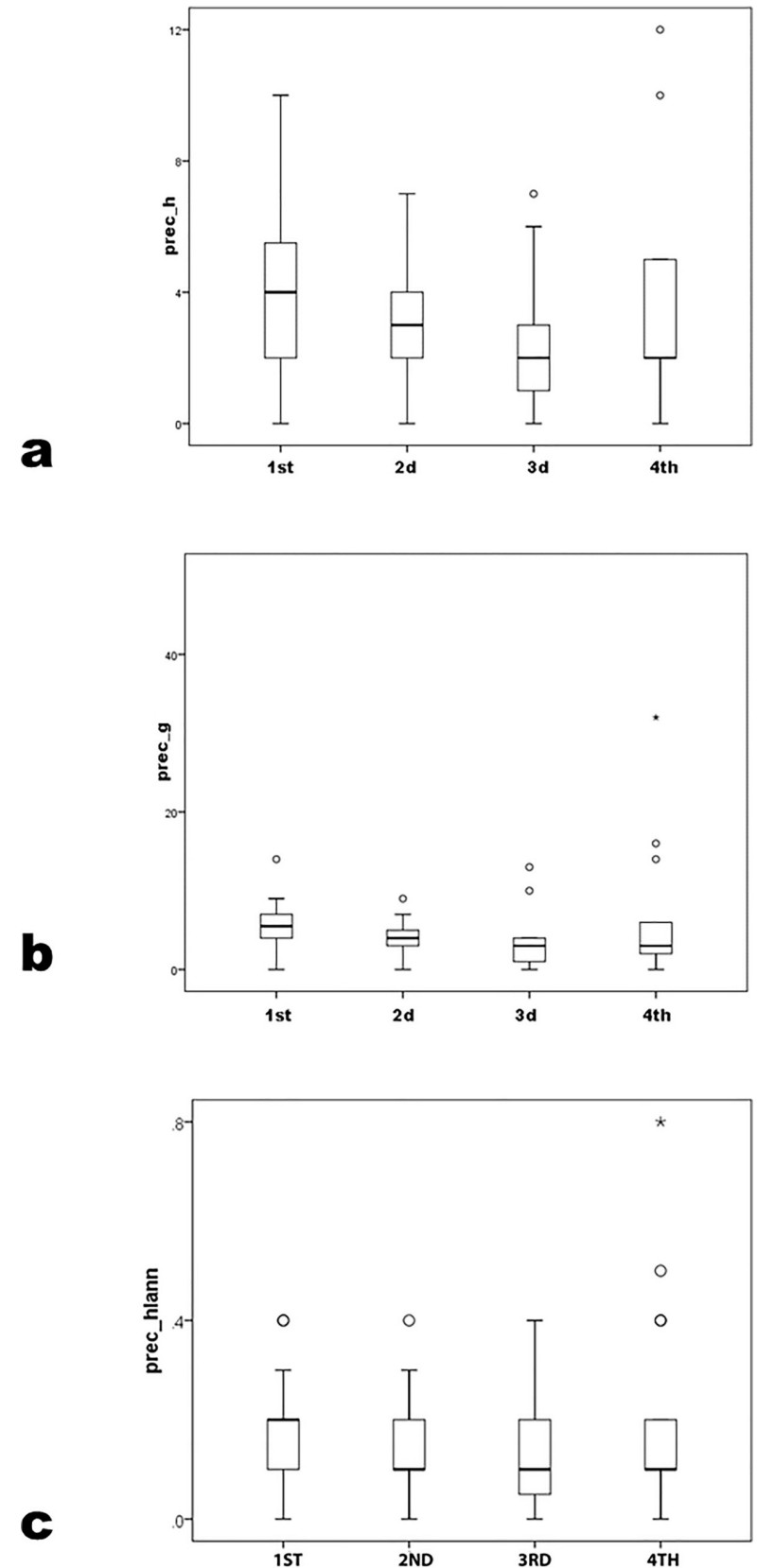

**Fig 10. prec_*h*, prec_*g* and prec_*hIann* by NRC-rank quartiles.**

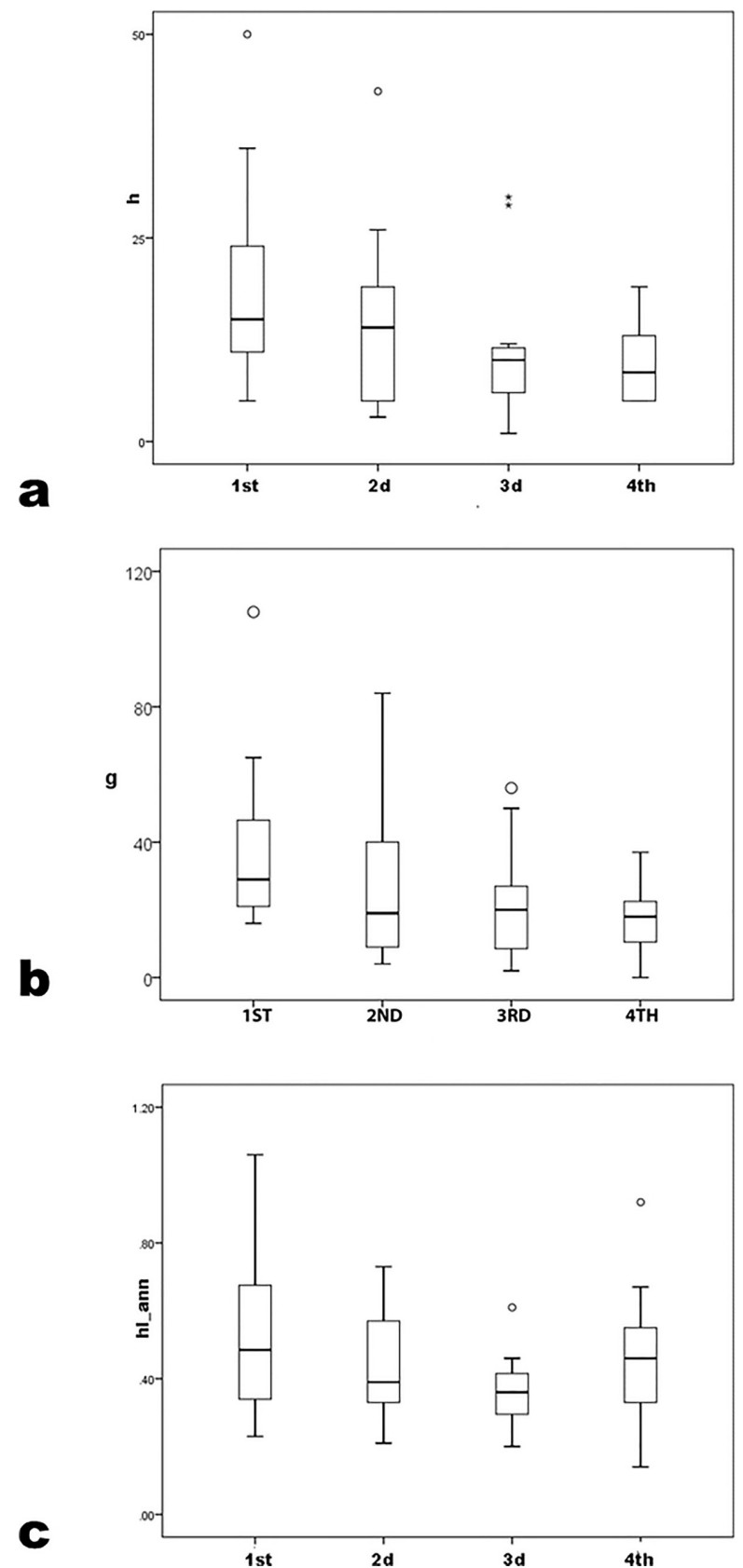

**Fig 11. *h*, *g* and *hIann* by NRC-rank quartiles.**

difference yields a conservative estimate of 100 new archaeology PhDs per year, or 2,000 over the comparison period. The 580 new hires comprise about 29% of that total, somewhat above Speakman et al.'s 2018 estimates. Some archaeology PhDs enter consulting practice or government agencies, and some do not seek academic careers. Yet these rough calculations suggest considerable overproduction of archaeology PhDs relative to academic demand which, coupled with the common practice of hiring new PhDs rather than mid-career archaeologists, itself suggests limited prospects for mid-career mobility.

Of the 44 archaeologists who appeared on faculty lists in both the 1997 and 2017 surveys, 35 (79%) remained at the same institution in 2017. Of the remaining nine, only four had moved to an institution possessing a higher degree program. One moved from an Ivy League BA program to a PhD program elsewhere, and one evidently was a trailing spouse. That leaves two of 44 (1.5%) who rose independently from a lower to an unequivocally higher placement. Movement at this rate amounts to rare exceptions who prove the rule. Mobility and the recruitment to higher placement that it might produce is rare in academic archaeology. Universalism's third corollary is not supported.

## Meritocracy test

Placement of individuals across program levels by merit quartiles (using *h* and *hIann* only) yields a matrix of affiliation for testing a status-mobility hypothesis of meritocratic allocation [50]. *h* being cumulative, the test was confined to the senior cohort of 61 archaeologists who earned the PhD in 1997 or earlier (Table 6) allocated among BA, MA, and lower and upper halves of NRC's PhD program ranks. Per [50], it rescaled proportions (rounded fractional values found in random-allocation matrices) to cohorts of 1,000 (by multiplying each cell by 1000/61 = 16.393), and compared the observed allocation matrix to random and ideal meritocratic ones. The measure is an index of dissimilarity that supports meritocracy if its value is less than that obtained from comparison of observed to random allocation.

Both merit measures return lower dissimilarity indices for the observed with the random than meritocratic matrix, i.e. the observed-placement matrix is more similar to the random, not meritocratic, one. Pooling placement groups in view of low original counts in 4x4 matrices does not alter statistical results.

The test treats scholarly merit as sole cause of placement, assumes that all archaeologists aspire to high placement, and therefore presumes strict meritocracy allocation. The real world is complex, but results indicate greater similarity of observed to random, not meritocratic, allocation, so fail to support a meritocratic hypothesis. They also suggest considerable overlap in program level by merit more consistent with random than meritocratic placement.

## Discussion

This study extends recent research on faculty placement and its determinants [7, 9], using recently developed databases and bibliometric measures that test propositions in the sociology of science. Precocity productivity does not pattern clearly by program level or rank of hiring institution, yet career-long productivity does with considerable overlap between levels. The pattern is consistent across program levels and graded subsets of NRC-ranked PhD programs, when controlling for age or cohort effects. Yet the trimmed sample, representing the great middle of archaeologists, and the upper *h* quartile of BA/MA affiliates approach or match PhD-level productivity norms.

Universalism's first corollary—promise correlating with initial hire—is supported only by distinguishing BA from higher degree programs, and then only modestly. Its second corollary —clear productivity sorting between levels—is not. The trimmed sample shows equivalent

productivity across levels. Mid-career mobility is extremely uncommon, casting doubt on universalism's third corollary. Only the "unproven assumption" ([4], 244) that hiring is guided by universalism supports the universalist view. Otherwise, data are more consistent with particularism [2, 11, 14]. If in higher education "the prize is prestige and the context is research productivity" ([22], 21), then placement in academic archaeology does not obviously function by meritocratic norms (see also [32] for sociology PhD-program publication counts, whose results clearly challenge a strict view of cumulative advantage that practically would confine scholarly productivity to higher-ranked departments, but did not examine this study's focus: productivity's possible correlation with program level).

Yet results engage the causal-inference problem noted above, individual productivity as either cause or consequence of cumulative advantage. No single study can solve the problem, but this one resembles others in suggesting particularism. For instance, a natural experiment in the cumulative effect of differential treatment in research awards among statistically undifferentiated young scientists concluded that cumulative advantage "may undermine meritocracy by allowing an initially fortunate scientist's recognition to self-perpetuate, while an equally talented but initially less fortunate counterpart remains underappreciated" ([51], 1; see also [20]). Following this logic, different sets of scholars can be practically identical in productivity measures at career start. Particularism is implicated if they then diverge in long-term productivity in ways not predictable from starting productivity but that correlate with factors like institutional support. That describes patterning in [51] and here.

Single cases are provisional, and these results beg their own questions for future research. Even if cumulative advantage explains the higher career—not precocity—productivity of the well placed, it may not do so alone. We need fine-grained data on variation in institutional support by degree program and by NRC rank groups, including on teaching loads and assistance, funds for research travel, special analyses, consultation and other purposes. Teaching loads likely are lower and institutional support higher in PhD versus lower degree programs. Whether they vary in degree corresponding to productivity differences among NRC rank groups should be determined. If resources that underwrite research vary systematically by program level and/or NRC rank groups, then cumulative advantage via institutional support is implicated. If, however, there are no significant differences in teaching load or other forms of institutional support among NRC rank groups, then their productivity differences must owe to nonmaterial factors or the meritocratic universalism that this study does not support.

We need more research to distinguish universalist and particularist causes. The former requires the supporting evidence the latter received here and elsewhere [2, 11, 52]. We must study possible halo effects [24] that may influence initial hire, acceptance of papers, or award of research grants independently of quality or quantity of scholarly production, perhaps using the intensive ethnographic methods that documented strong halo effects in fellowship awards [53], and access to journal and foundation archival data. We also must study hiring networks [7, 9] and gauge the size and connectivity of research and citation networks. Post-doctoral fellowships are not particularly abundant in archaeology, but if fellows are recruited as permanent hires disproportionately to PhD programs, the obvious productivity advantage they provide also should be considered and the role of merit versus particularism in their award investigated.

Tenure may be granted at lower rates in PhD versus lower degree programs, and/or in higher NRC rank groups among PhD programs, although a recent study of computer science departments found no significant difference in tenure rates across its prestige hierarchy ([20], 73). Such practices would have the effect of retaining only more productive scholars which, all else equal, would raise PhD programs' productivity profiles compared to other groups. Answering this question requires longitudinal data on hiring and tenure-granting rates.

Resource differences and halo effects both implicate cumulative advantage, yet identify distinct dimensions whose relative effects remain undetermined. Possible higher selectivity in the award of tenure is evidence of neither cumulative advantage nor universalism. Instead, it is a quality-control factor that, no matter how it harms both those who fail to earn tenure in some programs and those who could meet those standards but had no opportunity to, promotes neither universalism or particularism.

This study and others suggest that cumulative advantage generates particularism and therefore serious placement inequities. If so, the question of remedies arises. Although well beyond this study's scope, at least it is worth suggesting that large state-university systems—California's comes to mind—might centralize their hiring and undertake periodic post-tenure evaluation. Scholars could earn tenure in the system, either the UC system alone or combined with the California State University system—but then be subject to transfer within the system as their scholarly careers develop to different degrees. This would preserve tenure while permitting merit-based mobility. Although it also would entail dislocation as faculty are reassigned according to their demonstrated merit, it would reward the meritorious in ways that are practically foreclosed today.

## Conclusion

If the study of individual scholars' placement has not been common in archaeology compared to other fields [54], it may grow more popular as efficiency of data collection improves [55]. In future comparisons, bibliometric measures for those in PhD vs other degree programs should be handicapped or weighted in proportion to the cumulative advantage they confer.

In the American archaeological professoriate, productivity patterns ambiguously by degree-program level at initial hire; later productivity patterns broadly by degree program but with extensive overlap. Most precocity measures do not differ between MA and PhD programs or, among the latter, by NRC subgroup. Even in BA programs, some archaeologists have high precocity scores and compare favorably in career productivity with those in higher degree programs. Yet career measures consistently sort aggregates of archaeologists across degree program, even between MA and PhD programs, and within ranked subgroups of the latter; differences fade in the trimmed sample that represents the majority of archaeologists. Disparities do not owe to measurable differences in initial promise nor, as above, to data distributions, mobility, or differences in seniority or academic rank across program levels. Instead, they imply cumulative advantage. By its nature, particularism as its cause cannot be measured directly or easily, but it is a parsimonious explanation for differences that are negligible at initial hire but deepen as careers mature in a field that effectively lacks mobility.

## Supporting information

**S1 File.**
(DOCX)

**S2 File.**
(DOCX)

**S1 Table.**
(XLSX)

## Acknowledgments

Paul Allison, Mary Frank Fox, Anne-Wil Harzing, James O'Connell, Anthony Olejniczak and Alexander Petersen read parts of the text and provided comments. I am responsible for any errors. I declare no competing interests.

## Author Contributions

**Conceptualization:** Michael J. Shott.

**Data curation:** Michael J. Shott.

**Formal analysis:** Michael J. Shott.

**Investigation:** Michael J. Shott.

**Methodology:** Michael J. Shott.

**Validation:** Michael J. Shott.

**Visualization:** Michael J. Shott.

**Writing – original draft:** Michael J. Shott.

**Writing – review & editing:** Michael J. Shott.

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
