## [Editor Report · Decision Letter 0]

28 Jul 2021

PONE-D-21-11471

Merit and placement in the American faculty hierarchy: Cumulative advantage in archaeology

Dear Dr. Shott,

The above-mentioned manuscript has now been withdrawn from the review process at PLOS ONE.

If this was a mistake, please do reach out to us at plosone@plos.org. Otherwise, no further action is needed from you at this point.

Sincerely,

PLOS ONE

---

## [Author Response · Author response to Decision Letter 0]

3 Aug 2021

References to line numbers pertain to the revised version, PONE-D-17-44425R1,except where the re-revised ms. is indicated. 

OK, now we have two additional reviews, one of which (#5) is strongly positive. That reviewer has a few minor reservations (I too have concerns about any ranking, as the original and revised version [lines 259-64] stated). On several of those minor points:

1. Line 120-21’s personal interactions occurred in the past so were not observed whether or not they were observable but, yes, can be observed if we have observers at interview lectures, personal meetings, and dinner conversations.

2. Degree program as “not…an important student criterion”. Inserting “undergraduate” before “student” should clarify, or the reference to students can be omitted from the sentence. By this (minor) statement I meant that undergrads probably don’t care which or how many graduate programs their undergraduate school offers. Small point, addressed by a small change in the text.

3. Omitted “s” on line 220 noted, as is “than” for “that” on line 641 and the extra period on 1065.

4. Finally, if necessary I can cite the sociology source identified on criticism of NRC Sociology rankings.

By now, then, we have three strongly positive reviews, plus #1 and #6. I’ve already expressed my opinion of #1, supported by detailed empirical and logical argument. That leaves #6. 

Reviewer 6. Near both start and end #6 states his/her agreement with “the other reviewers” when in fact by now three of five have delivered strongly positive, slightly qualified reviews, none of which made remotely “extensive” comments about prose. No reviewer—not even #1—gave “extensive discussion” to writing style or clarity, although #1 did impugn it without substantiating the charge, this in a review whose text included words like “self-indulgery.” More accurately, then, #6 agrees with one, not all reviewers, although #6 is slightly less negative than is #1.

Much of this review is devoted to a detailed discussion of…the abstract, which it practically parses. Abstracts are not intended for lengthy discourse on definitions or to present an argument in detail. Merriam’s defines “abstract” as a “summary of points (as of a writing) usually presented in skeletal form” (original emphasis). Skeletal form. Yet #6 criticizes my abstract for failing to flesh out my argument in all details. Summaries—abstracts—by their nature cannot and should not attempt to present arguments in nearly all relevant detail. On the contrary, good ones should state the problem or question, describe data compiled and analysis conducted to address the question, and state conclusions. All briefly. 

The use of “pattern” as a verb is well-established in academic English and requires no justification. (While I’m consulting Merriam’s, note that it too treats “pattern” as a verb. And Merriam’s defines it exactly as I used it.) The review of the abstract offers repeated comments about terms undefined there, all of which are defined in the text. #6 is entitled to disagree with me that abstracts should define terms. But let’s also consult PLoS One on the matter. Here is how the journal describes abstracts: “The Abstract should:

• Describe the main objective(s) of the study

• Explain how the study was done…without methodological detail

• Summarize the most important results and their significance

• Not exceed 300 words”

Seems reasonable to me, and consistent with my practice over the past 35 years. Yet #6 criticizes my abstract for failing to describe what, in this context, can only be called “methodological detail” and not merely summarizing, but documenting in detail, its “most important results and their significance”. In effect, #6 criticizes the abstract for conforming to PLoS One editorial standards. Who is s/he arguing with? 

On #6’s specific comments about my abstract:

• I fail to make clear “the distinction between MA and PhD-granting programs.” Ms. lines 198-200, among others, cover this point. #6 then accuses me of poor writing because the distinction’s salience “only becomes clear later in the paper…” Yes, in the paper. Not in the abstract. This is an absurd criticism: I fail to make my full argument in an abstract that PLoS One itself agrees is not to be used for that purpose.

• The abstract’s third sentence fails to define “ranked subsets of PhD programs,” a failure that #6 notes twice. But of course I didn’t define those subsets there; the abstract isn’t the place for the definition. Rev. #6 should read ms. lines 927-978, a section that treated ranked subsets of PhD programs, where the definitions appeared on lines 935-36 and 955-56.

• The next comment demands insertion of a comma two words after an existing comma. I’ve reworded, slightly and differently, to clarify what I honestly believe most readers would understand clearly.

• Abstract-parsing continues by professing bewilderment at what “mobility” may mean. Again, recalling that PLoS One urges brevity and summary, not detailed disquisition, I can only refer #6 to the actual text. Mobility is defined on ms. lines 834-40, especially 839-40. It all seemed reasonably clear to me, but re-revision rewords further. At the same time, I welcome any specific, constructive suggestion to make the passage even more clear.

• #6 writes “One can only guess” about a supposedly unclear point, also related to mobility. One should read the text before making such comments, in this case lines 838-40. 

• The abstract’s sixth sentence states a conclusion that #6 doesn’t like. But #6 doesn’t admit as much, instead suggesting that the difference in averages by program level is inconsistent with the conclusion. That is a distortion of my argument and, for that matter, Long’s and Merton’s before me. It focuses exclusively upon career measures, and deliberately avoids their inconsistent patterning with precocity ones which, of course, is a key component of the analysis.

• #6 then concludes that “the abstract already took me a good deal of time to understand correctly.” Two things. First, #6 does not understand the abstract or the text correctly, despite whatever time invested, as indicated by the plain reference to text passages that explain what s/he condemned the abstract for failing to describe or document in detail (i.e., criticizing the abstract for what it cannot be). Nor does s/he take it at face value, as indicated by the distortion of my logic and conclusions noted above. None of #6’s failings here owe to any expository flaws of the abstract or text, merely to any a priori views that s/he holds and to her/his treatment of abstracts as complete arguments. Second, sincerely I’m sorry that it took Rev 6 “a good deal” of time to read the abstract; it took me probably more time tediously to refute his/her unsubstantiated criticisms and respond to his/her charges that the abstract must do what only the text may do. PLoS One should monitor its reviewers’ conduct so that authors are not required to respond to tedious and ungrounded “criticisms.”

Yours to me of 20 August expressed “concerns about the readability of the manuscript,” which I take as an implicit requirement to improve that “readability.” Apart from minor comments that three reviewers made about wording, to which I’ve responded, the “concerns” were expressed by #1 and #6. As above, #1’s criticisms were severe; recall that s/he rated the paper “incoherent.” Alas, those criticisms weren’t merely not well-documented; they weren’t documented at all. My earlier response dealt directly with #1’s unfounded, objections. Now, #6. I discount #6’s parsing of the abstract as an attempt to criticize my writing by failing to understand the purpose of abstracts. I am not an exceptional writer, merely somewhat above average compared to the standards of my field and others I’ve read. Criticisms so broad, vague and undocumented are no basis for “improving readability” of a perfectly readable ms. They are ungrounded accusations. If there were significant, clearly explained “concerns,” I would respond to them. 

I am entitled to better documentation of my prose flaws than the questionable parsing of an abstract that presumes that abstracts must present the totality of arguments, including definition of terms. #1 and #6 were incapable of documenting the broad charges they made. 

On #6’s substantive comments. #6 may repeat #1’s error about precocity measures. Those measures do not necessarily “reduce” to number of publications (P), but differences are apt to be slight for young scholars. That’s fair enough. #6 then states that there is a “clear correlation” between measures and program level in career measures, also correct. The correlation is in fact statistically significant although #6 might have noted, as the ms. does (lines 785-790), the considerable overlap in measures between program levels, and between NRC sub-groups. Even the abstract made that point (lines 43-5). Yes, there’s a pattern. But there’s sufficient diffusion in it to refute the argument—implicit in #6’s comments—that there is strong correspondence.

Now on #6’s “three objections” to the ms.’s qualified conclusion.

His/her Point 1.

Like #1, #6 reduces merit--I presume s/he means precocity merit—to P. Let’s be as clear here as I feel an unbiased reader of the ms. would be. In both precocity interval and over the entire career, all analysis is conducted using cited bibliometric measures, not P. (Table 1 lists mean publications by category, for comparison only.) So no, merit is not “now” or at any other time defined as P. This is exasperating. The study design requires productivity measures that pertain to the period of initial hire. If #6—or #1 previously—had constructive suggestions about other, presumably better, such measures or for study redesign, I’d give them serious regard. Instead, they merely point out the same limitation of those measures that the ms. itself (lines 606-07; don’t recall line #s in the original version, but the same acknowledgment was there) freely acknowledged, as though it were both an original observation and a damning flaw of the study. Not constructive.

First, I must repeat what ms. lines 612-17 already stated, and whose purpose and point I would have thought were clear. Precocity measures correlate strongly with one another, and career measures correlate strongly with one another. However, precocity h correlates at much lower r (and corresponding r2) values with career h, as do other precocity measures with their career equivalents. There also is considerable overlap in the range of career values by program level. That is, precocity measures provide comparatively little basis for predicting career performance. The re-revised ms. underscores this point. Reluctantly, I also attach text and figures that could be added to the re-revised ms. after line 583, although I hope it wouldn’t be necessary.

#6 then freely speculates in support of his/her universal ideal, in ways strikingly reminiscent of the Coles’ views summarized in ms. lines 107-124, and 149-150. Note line 112’s “’unproven assumption’”; these are not my words but the Coles’ own. The best account they could give of their objection to any suggestion of particularism was an assumption! I belabor this small point only because #6 invokes similar assumptions. S/he considers it “very reasonable to suppose” these shrewd judgments that bibliometric measures cannot reveal (of course not) but that must have occurred. On the strength of logic of such quality, #6 declares my conclusion a “far stretch.” It is no farther, a good deal shorter, stretch than to invent “reasonable” suppositions to the contrary, without evidence to support them. At least I have a considerable body of evidence to support my view. Of course reviewers have no obligation to make an ms.’s case for it, but their criticisms must be based on something better than supposing, without a shred of supporting evidence, that “quality, creativity or novelty” win out. Nor does such casual criticism therefore refute a considerable body of evidence that supports a reasoned conclusion. That’s special pleading, not serious argument or fair criticism.

Ms. lines 117-124 (and 160-61) anticipated #6’s critical ploy, so essentially I’ve already replied to this objection. Please read them carefully. Then please tell me what argument of any kind or quality cannot be disputed by “reasonable suppositions” to the contrary? #6 is entitled to dispute, but to be considered fair and taken seriously the dispute must be grounded, not a matter of free supposition. Special pleading is an all-purpose bludgeon, against which there is no defense. That isn’t because it is valid, but precisely because it cannot itself be disputed on either empirical or logical grounds. Yes, it’s always possible that there may be some other factor to explain a pattern, about which free supposition can freely suppose. Only grounded criticisms deserve responses, and they do not rely upon poor arguments. This review is ad hoc, not grounded or reasoned.

It’s a small point in this context but, with respect to whatever #6’s field is, “geography or methodology” are in fact highly relevant hiring criteria in archaeology. If #6 disagrees, s/he should read the last 30+ years’ position announcements in archaeology. #6 also might have noted ms. line 1089, which cited an earlier study I wrote that clearly demonstrated exactly the pattern in geographic preference that s/he dismisses. I don’t criticize #6 for not knowing about archaeology’s clear geographic biases of long standing, but am astonished at the presumption that s/he knows the field and that such bias does not exist. 

His/her Point 2.

I’m uncertain what is meant, but intrigued by one possibility. #6 “reconsiders” the documented absence of correlation between productivity and placement in the precocity interval (note that s/he takes at face value the equally documented correlation in later career) because the ms. “effectively selects on the longevity of the career.” By this #6 evidently suggests a difference in career length between program levels that is hidden or overlooked in the study and that explains the pattern in precocity values. The evidence to support such a charge? None. Thus, #6 supposes that some would have left academia; in a population of this size, about 875, that’s likely. Fair enough, again. How those departures are distributed in time or by program level is what’s at issue. To that degree, #6’s argument is fair if one: 1) supposes that those who left had low precocity scores; 2) supposes that they were hired by low-ranked universities, and; 3) supposes that they left because they failed to meet even the presumably low standards of those universities. If all of these suppositions hold then yes, there is bias “inherit [sic] to the design of the study”. (Knowing the challenge of writing in a nonnative language, I don’t ordinarily criticize the English prose of nonnative speakers. But #6 presumes to write English well enough to lecture me on how to write it.) Again, this is supposition, in this case more involved even than invoking the mere supposition that novelty, etc., are rewarded at hire. I can’t disprove it any more than I—or even #6, come to think of it--can prove a negative.

But consider that if #6’s scenario holds, then there should be significant difference in faculty proportions by professorial rank. If poor scholars are being hired (unwisely) by lower-ranked universities that later experience a form of buyer’s remorse by (presumably wisely, if ruefully) refusing them tenure, then faculty in lower program levels are being replaced more often. Therefore, that part of my sample should be skewed toward lower professorial rank, higher program levels toward higher professorial ranks. No such pattern is found in my data, as ms. lines 704-15 reported. On the contrary, proportions of professorial rank by program level are similar. I have evidence, not supposition, to support this conclusion. Thus, the study does not, as #6 charges, “select upon the longevity of career,” because career length does not differ by program level or NRC subsample. The re-revised ms. makes that point against #6’s supposition. Actually, against #6 I might suppose that the process s/he describes might play out more often in PhD programs. If so, then those, not lower, programs should skew toward younger faculty and lower professorial ranks. Again, however, there is no such pattern in the evidence. Reluctantly, I am forced to add a new section (re-revised ms. lines 794-819) merely to refute this dubious argument. I confess that the logic it sets forth is tortured, but only because it mirrors #6’s supposition in that respect.

Finally on this point, #6 again sees ambiguity, this time in what I mean by “mobility.” This point merely repeats his/her criticism of the abstract, addressed above. 

Point 3.

Point 3 reduces to #6’s Point 1. Once more, there is free supposition. I do not necessarily agree that capabilities are any more poorly measured in the precocity interval than the later career one. I agree that those talents are measured “by long-term productivity” only if controlling for the effects not just of time but the setting—here, program level—in which that time is spent. Indeed, that is the heart of the matter. If we take productivity measures at face value, without considering the conditions under which academics labor in order to earn them, then we elevate universalism to unassailable article of faith. That’s like taking annual income at face value as a measure of merit without distinguishing between we who earn our own and the fortunate few who inherited fortunes. At that point we can, as #6 does, argue by predilection rather than be evidence. If by such dubious logic #6 means that savvy hiring committees see the je ne sais quoi that precocity measures fail to measure but that career measures can (again, #6 is quite happy to accept, without reflection or examination, a correlation b/w career measures and program level that supports his/her a priori view, but labors mightly to suppose this or that against a correlation that opposes it), then I must request an explanation. How is it that career measures capture that ineffable essence while precocity measures do not? You can’t have it both ways. If you like bibliometric measures over the entire career, you can’t dismiss those that are compiled over the precocity interval, nor how they pattern relative to career measures, merely because they are inconvenient to your preconceptions. 

Lines 113-24 address objections the nature of #6’s, and before s/he even expressed them. I must call to #6’s attention something already stated in the ms.: “Invoking such unobservable factors against a particularist conclusion at once renders it difficult to prove and its universalist alternative difficult to refute.”

Trace out #6’s logic. If the evidence supports universalism, then the universalist conclusion must be reached. If the evidence opposes it, then the universalist conclusion still must be reached, in this case by free supposition that invents convenient causes for the patterns in evidence that are inconsistent with it. This is not merely transparently unfair but a tautology. However the evidence patterns or free supposition claims, it supports #6’s predetermined conclusion.

In #6’s mind, what evidence is necessary to support a qualified conclusion of particularism? His/her standards are highly unpersuasive, returning us to nothing better than the Coles’ “’real-life experiments…difficult to perform’” (ms. lines 110-11). Difficult indeed; nothing short of impossible, as the Coles themselves suggested.

Against #6.

I document a clear pattern: no productivity differences in the precocity interval, but emerging in later career. Then, following a considerable body of earlier research that ms. lines 139-140 cite, I suggest a cause: higher support for research combined with Matthew effects that benefit those hired into PhD programs. The suggested causes correspond to the pattern documented. These are factors the nature of which Merton cited (and, as my text and I’m sure others demonstrate, so have many subsequent scholars) as sources of particularism. Therefore, I believe that the result is both documented and significant, and suggest that it is consistent with particularism. To rescue his/her contrary logic, #6 cites no—none, not an iota of--supporting literature or evidence, and instead merely indulges in free supposition. 

Mindful of ms. lines 107-124, I am entitled to turn #6’s logic—and #1’s—against them:

1. If hiring were truly meritorious, we should see a clear difference in precocity scores by program level at placement. We do not.

2. If hiring were truly meritorious, we should see a strong correlation between bibliometric measures in precocity and career-long intervals. We do not.

3. If hiring were truly meritorious, we should see little if any overlap between program levels that is visible in career productivity measures. Instead, considerable overlap is evident, as the abstract--which #6 parsed laboriously on editorial grounds but did not acknowledge--clearly states.

4. Anyone who feels free to suppose freely to the contrary has commensurate obligations. One option is to argue by free supposition as did #6, but only acknowledging that his/her objections are speculative, not grounded. Another is to offer either evidence, not mere supposition, to the contrary, or at least to suggest positive, constructive ways independently to test at once one’s own suppositions and perhaps contrary views. It has been years, but I once read Popper. How, in the least way, is #6’s contrary view falsifiable? If it is not, then how does it deserve serious regard, and what defense does any legitimate argument like mine have against it? Anyone can say “Yes, but….” #6 must prove that I’m wrong, or at least offer grounded, perhaps even constructive, criticisms, not merely freely suppose that I might be wrong, and thereby further suppose that s/he refuted my argument or even so much as offered constructive criticisms. The currency of academic criticism is documented, fair, argument and evidence, not free supposition.

Beyond the ms.’s empirical and analytical contributions, lines 986-1013 and 1105-26 qualify results, suggest other kinds of evidence that might support or refute my tentative conclusion, and advocate research to collect and analyze such evidence. They demonstrate my willingness both to be proven wrong and to constructively suggest lines of research, not to freely suppose in armchair fashion.

Free supposition is no basis for a fair professional review.

---

## [Editor Report · Decision Letter 1]

12 Oct 2021

Merit and placement in the American faculty hierarchy: Cumulative advantage in archaeology

PONE-D-21-11471R1

Dear Dr. Shott,

We’re pleased to inform you that your manuscript has been judged scientifically suitable for publication and will be formally accepted for publication once it meets all outstanding technical requirements.

Kind regards,

Radu Iovita

Academic Editor

PLOS ONE

Additional Editor Comments (optional):

After discussions with the Chief Editor, we came to the agreement that your revisions and rebuttals of the reviewer comments were sufficient to guarantee that your submission fulfilled the publication criteria without sending it out for another round of reviews. We also agreed that, should others find your article or its findings objectionable, these objections should take the form of a separate article, rather than be lost in the process of peer review. 
---

## [Editor Report · Acceptance letter]

20 Oct 2021

PONE-D-21-11471R1 

Merit and placement in the American faculty hierarchy: Cumulative advantage in archaeology 

Dear Dr. Shott:

I'm pleased to inform you that your manuscript has been deemed suitable for publication in PLOS ONE. Congratulations! Your manuscript is now with our production department. 

Kind regards, 

on behalf of

Dr. Radu Iovita 

Academic Editor

PLOS ONE